# A thermostable type I-B CRISPR-Cas system for orthogonal and multiplexed genetic engineering

Zhiheng Yang [1,2,5], Zilong Li[2,5], Bixiao Li [2,5], Ruihong Bu[2,3], Gao-Yi Tan [1], Zhengduo Wang[1], Hao Yan[2], Zhenguo Xin[2], Guojian Zhang[3], Ming Li [2,4], Hua Xiang [2,4], Lixin Zhang[1] ✉ & Weishan Wang [2,4] ✉

Thermophilic cell factories have remarkably broad potential for industrial applications, but are limited by a lack of genetic manipulation tools and recalcitrance to transformation. Here, we identify a thermophilic type I-B CRISPR-Cas system from *Parageobacillus thermoglucosidasius* and find it displays highly efficient transcriptional repression or DNA cleavage activity that can be switched by adjusting crRNA length to less than or greater than 26 bp, respectively, without ablating Cas3 nuclease. We then develop an orthogonal tool for genome editing and transcriptional repression using this type I-B system in both thermophile and mesophile hosts. Empowered by this tool, we design a strategy to screen the genome-scale targets involved in transformation efficiency and established dynamically controlled supercompetent *P. thermoglucosidasius* cells with high efficiency (~ $10^8$ CFU/μg DNA) by temporal multiplexed repression. We also demonstrate the construction of thermophilic riboflavin cell factory with hitherto highest titers in high temperature fermentation by genome-scale identification and combinatorial manipulation of multiple targets. This work enables diverse high-efficiency genetic manipulation in *P. thermoglucosidasius* and facilitates the engineering of thermophilic cell factories.

Microorganisms are regarded as excellent cell factory for the production of a large diversity of chemicals in industry[1–4]. Despite such achievements have been made, some challenges still remain, including low mass transfer efficiency, expensive and knotty bioreactor cooling, and frequent mesophile contaminations[5]. Therefore, it is important to find new methods to further improve competitiveness of microbial cell factory. For this, thermophilic microorganism producers give us an attractive solution[6]. High-temperature fermentation, closer to the temperature used in chemical refineries, offers a number of advantages over mesophilic biorefineries[7], including faster feedstock conversion rates, reduced bioreactor cooling costs, lower risk of contamination, and decreased viscosity during fermentation[6,8].

However, due to insufficient genetic manipulation and the severe lack of well-developed genome-editing tools, the development of such thermophilic microbes dramatically lags behind that of mesophiles such as *Escherichia coli* or *Saccharomyces cerevisiae*[2,9]. In particular, thermophilic *Parageobacillus thermoglucosidasius* (previously *Geobacillus thermoglucosidasius*), has emerged as a promising industrial workhorse with fast growth at 55–65 °C and the ability to use a broad range of carbon sources including monosaccharides, cellobiose, and short-chain oligosaccharides[10–12]. Although *P. thermoglucosidasius* has been used to produce several high-value chemicals[13–16], alternative energy sources[17], and enzymes[18], its potential is largely untapped due to a lack of tools for high-efficiency and versatile genetic manipulation.

[1]State Key Laboratory of Bioreactor Engineering, and School of Biotechnology, East China University of Science and Technology (ECUST), 200237 Shanghai, China. [2]State Key Laboratory of Microbial Resources, Institute of Microbiology, Chinese Academy of Sciences, 100101 Beijing, China. [3]School of Medicine and Pharmacy, Ocean University of China, 266003 Qingdao, China. [4]University of Chinese Academy of Sciences, 100049 Beijing, China. [5]These authors contributed equally: Zhiheng Yang, Zilong Li, Bixiao Li. ✉e-mail: lxzhang@ecust.edu.cn; wangws@im.ac.cn

Clustered regularly interspaced short palindromic repeats (CRISPR) and the associated CRISPR-associated (Cas) proteins endow prokaryotes with adaptive and heritable immunity. Not only serving as guardians against invading genetic elements, these systems have been engineered as genetic tools that revolutionized the genetic manipulations[19]. Based on the components and characteristic proteins, the CRISPR-Cas systems are classified two classes (Class 1 and Class 2) and six types (Type I to Type VI)[20]. In contrast to the widely used single-Cas proteins (Class 2), a large number of unexploited type I CRISPR-Cas systems (Class 1) are is widely distributed in prokaryotic genomes. Currently, endogenous type I CRISPR-Cas systems can be deployed for genome cleavage-based editing or binding-based transcription regulation, dependent on the presence of intact Cas proteins or deletion/inactivation of Cas3 nuclease, respectively[21,22]. Distinct from this mode, Li et al. recently reported a regulatory mechanism of type I-B systems in *Haloarcula hispanica* in which Cas effectors can be guided by non-canonical crRNAs to mediate transcriptional repression without ablating Cas3 nuclease[23]. They further developed a tool for simultaneous gene regulation and editing in this host[24]. However, these advances were conducted in a halophilic organism with high salinity requirements for activity and stability that are prohibitive to the large majority of microorganisms. We therefore sought to characterize and harness an ideal thermostable type I-B CRISPR-Cas system to address current limitations in the efficiency of genetic manipulation tools for engineering thermophilic microbial factories.

Concurrent with developing a versatile genetic manipulation tool with orthogonal genome editing and transcriptional repression, we further seek to identify and reprogram the genes underpinning low-transformation rates in *P. thermoglucosidasius*[25], since efficient transformation is essential for basic research of biological and physiological processes as well as for engineering strains for specific applications. Supercompetent cells were previously generated in *E. coli* DH5α for general cloning applications by deleting or disrupting multiple genes related to DNA uptake and replication, such as *recA1*, *endA1*, *deoR*, or *phoA*[26]. However, static, functional inactivation of these conditional essential genes compromises the health of DH5α cells, resulting in their poor suitability for industrial fermentation conditions and limiting their application in large-scale production operations[27,28]. To improve transformation efficiency without impairing cell health, the genes participating in transformation should be identified at genome scale and reprogrammed in a multiplexed temporal control manner.

Here, we discover a thermostable type I-B CRISPR-Cas from *P. thermoglucosidasius*, and find that it can cleave target DNA using full-length complementary crRNAs, or repress transcription when guided by truncated complementary crRNAs (< 26-nt). Following identification of its exact protospacer adjacent motif (PAM) sequence, we develop a plug-and-play module with this type I-B CRISPR-Cas for orthogonal genome editing and transcriptional repression in both thermophile and mesophile chassis cells with active nuclease Cas3. After showing the high efficiency of this tool through a series of simultaneous genetic manipulations, we then provide a proof-of-concept demonstration of its power by overcoming low-transformation efficiency in *P. thermoglucosidasius* through (1) the development of a screening strategy for genome-wide identification of genes related to transformation efficiency, and (2) the design of a multi-gene regulatory module to dynamically control the identified target genes. This process results in temporally inducible super-competent cells with a high transformation efficiency of ~$10^8$ colony-forming units (CFU)/μg DNA. This design guarantees that the growth and metabolic function is unaffected by the multiplex gene manipulation when the module is in the inactive state (i.e., without the xylose inducer). Facilitated by abovementioned work, we further demonstrate the construction of thermophilic riboflavin cell factory with hitherto highest titers in high-temperature fermentation by genome-scale identification and combinatorial manipulation of multiple

targets. The type I-B CRISPR-Cas orthogonal genetic manipulation tool reported here provides a versatile and scalable approach to functional genomic screens and engineering thermophilic strains for a wide range of industrial applications.

## Results

### A thermostable type I-B CRISPR-Cas system from *P. thermoglucosidasius*

*Geobacillus* and *Parageobacillus* are ubiquitous in a wide range of environments, growing at temperatures ranging from 35–80 °C[29]. To survey the distribution of thermostable type I CRISPR-Cas systems in these genera, we extracted all available genomes from CRISPRCas-Finder database[30], and identified Type I-B as the most prevalent CRISPR-Cas system (Supplementary Fig. 1). Considering the application potential, we selected the I-B CRISPR-Cas from *P. thermoglucosidasius* NCIMB 11955 for further investigation. It comprises eight Cas proteins (Cas1–7 and Cas8b) and three CRISPR arrays (CRISPR1, CRISPR2, and CRISPR3) harboring 28, 33, and 34 spacers, respectively (Fig. 1a and Supplementary Data 1). Downstream of CRISPR3, we also found a type III-B CRISPR-Cas system with four cognate arrays (with 30-nt repeats and 35–39-nt spacers) ~1.48 Mb from the type I-B CRISPR-Cas that had variable spacer sequences. To identify target protospacers, we employed the web CRISPRTarget[31], which found a match between CRISPR1 spacer18 and one protospacer from the plasmid pGEOTH01 (1 base mispair), between CRISPR2 spacer19 and DNA sequence of *Geobacillus* virus E2, and between CRISPR3 spacer32 and target protospacers in *Geobacillus* phage GBSV1 (Fig. 1b).

To characterize the activity and thermostability of the type I-B CRISPR-Cas system, we further examined our alignment, which revealed conserved repeat sequences between the type I-B and type III-B CRISPR arrays (Fig. 1c). To avoid potential cross-talk between the two CRISPR-Cas systems, we deleted all type III-B Cas proteins from strain NCIMB 11955 by our previous method[32], resulting in strain Y1 (Supplementary Fig. 2a). We then conducted reverse transcription PCR assays to confirm that all type I-B genes were indeed expressed in Y1 (Supplementary Fig. 2b). Subsequently, all three of the above protospacers, including the 5 bp upstream PAM sequence were then cloned into a control plasmid to generate target plasmids (pVirus, pPhage, and pGeoth, respectively) for interference assays. In these assays, an active native crRNA-Cas protein should cleave and degrade the target plasmids, resulting in the absence of clones on a selective medium. We found that few or no colonies were generated by Y1 transformed with the pVirus and pPhage targets, which shared 100% protospacer identity. As expected, we also observed that Y1 displayed slightly lower interference activity towards pGeoth (Fig. 1b, d), while correcting the 1-nt mismatch in the pGeoth protospacer (i.e., pGeoth*) resulted in high interference activity (Fig. 1d). These results indicated that this type I-B CRISPR-Cas system had a high capacity for targeting and interference activity.

Further, testing of the type I-B CRISPR-Cas system at different temperatures in native strain Y1 indicated that it could target the pVirus plasmid with ~100%, 99%, and 99% efficiency at 45 °C, 60 °C, and 70 °C, respectively (Fig. 1e). Since *P. thermoglucosidasius* cannot grow <42 °C, we transformed a plasmid harboring the type I-B Cas effector cascade and the single-spacer mini-CRISPR into *E. coli*, which showed an interference efficiency of 100% at 37 °C (Fig. 1f and Supplementary Fig. 3a, b). Collectively, these results demonstrated that this type I-B CRISPR-Cas system was active and thermostable in both thermophilic and mesophilic strains.

### crRNA length determines orthogonal transcriptional repression or cleavage activity

Inspired by the finding that noncanonical crRNAs (creA) can serve as guide RNAs to recruit cascade and subsequently repress the transcription of toxin gene (*creT*) by partial complementarity in *H. hispanica*[23], we sought to determine whether truncated crRNAs could

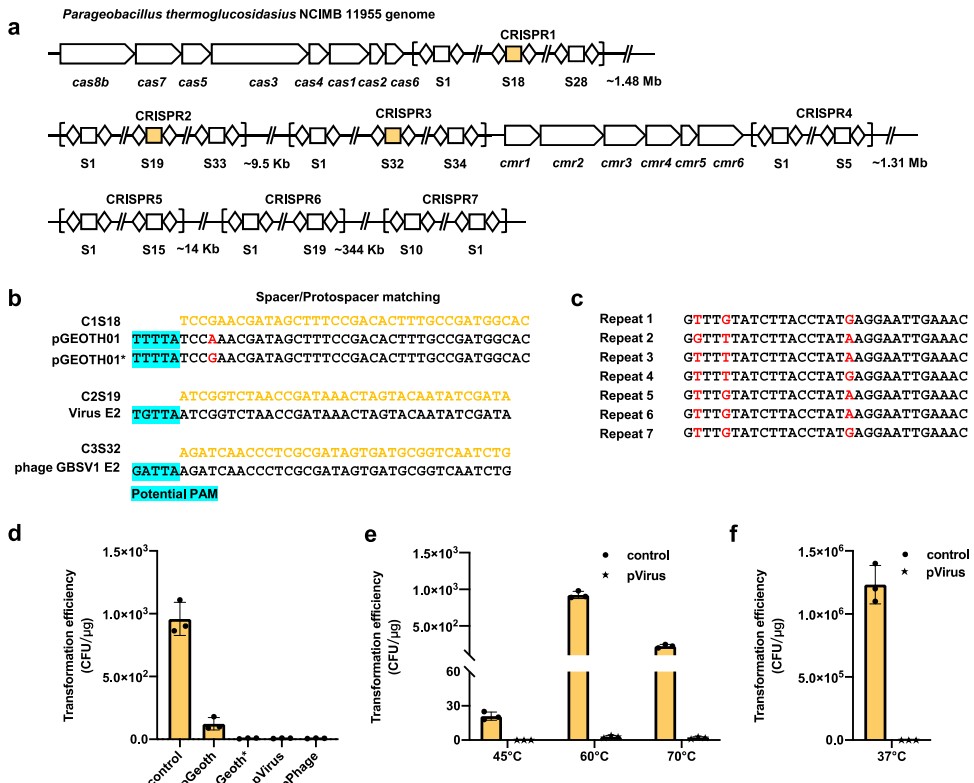

**Fig. 1 | Identification of the thermostable type I-B CRISPR-Cas from *P. thermoglucosidasius*. a** Genetic architecture of the Type I-B and III-B CRISPR-Cas operon. The diamond: repeat sequence. The square: spacer sequence. **b** Protospacers predicted by web CRISPRTarget tool. The yellow sequences: spacers from genome. The black sequences: protospacer sequence. The potential PAM is indicated by a blue background. **c** Sequence alignment of seven repeats from Type I-B and Type III-B CRISPR-Cas system. Non-conservative sites are highlighted in red. **d** Plasmid interference activity of the type I-B CRISPR-Cas system in strain NCIMB 11955. The transformation efficiency was indicated by the rate of recovered clones. A low-transformation efficiency indicates a high plasmid interference activity. **e** Plasmid interference activities of the type I-B CRISPR-Cas system at different temperatures in strain NCIMB 11955. **f** Plasmid interference activities of the type I-B CRISPR-Cas system at 37 °C in *E. coli*. Data represented three biological repeats and displayed as mean ± SD. Source data are provided as a Source Date file.

guide the transcriptional repression activity of type I-B CRISPR-Cas in *P. thermoglucosidasius*. To facilitate evaluation of transcriptional repression activity, we introduced a *sfgfp* reporter gene driven by the $P_{ldh}$ promoter into strain Y1 to produce strain Y2 (Fig. 2a). Monitoring fluorescence over time confirmed that *sfgfp* was constitutively expressed in Y2 (Supplementary Fig. 4). Based on its effectiveness as a PAM in the above plasmid interference assays (Fig. 1d–f), we searched the *sfgfp* sequence for the 5-nt upstream pGEOTH01 protospacer sequence and found "TTTTA" motif on the non-template strand (target 1). We therefore introduced truncated crRNAs targeting this motif just downstream of TTTTA sequence in strain Y2 by a repression plasmid expressing the crRNAs driven by the pRplsWT promoter[25] (Fig. 2a).

A series of progressively shorter crRNA truncation variants 30-nt, 25-nt, 20-nt, 15-nt, and 11-nt in length with 100% complementarity to *sfgfp* target (Fig. 2b). In contrast to the lethal effects of crRNAs with ≥30-nt complementarity (Fig. 2c), crRNAs ≤25-nt had no significant effect on the growth of strain Y2 (Supplementary Fig. 5a), indicating that crRNA truncation variants did not guide target cleavage. Further tests of their effects on transcriptional repression showed that fluorescent signal was inhibited to a great extent in cells expressing the truncation variant crRNAs (Fig. 2b, d), which was confirmed by qRT-PCR assays (Fig. 2e). These results indicated reducing the length of completely complementary crRNAs could orthogonally guide the Cascade proteins to perform the role of transcriptional repression.

To determine the point at which cleavage-based interference activity switches to transcriptional repression, we evaluated the interference and repression activity of 26–29 nt fully complementary crRNAs. These assays indicated that interference activity, but not repression, was lost in crRNAs with ≤26-nt complementary sequence (Fig. 2f, g). To further strengthen the evidence that 26-nt was the switch point, we constructed two derivative strains (strain T-26 and T-27) harboring an inducible mini-CRISPR array with a truncated spacer of 26 or 27-nt (Supplementary Fig. 5b). A plasmid containing the corresponding protospacer target was then introduced into strains T-26 and T-27. Monitoring plasmid loss at 4 h post-induction of crRNA expression in antibiotic-free medium revealed that the plasmid was lost in strain T-27, whereas no discernible change in plasmid number could be detected in strain T-26 after induction (Fig. 2h), indicating that the 27-nt crRNA could guide cleavage activity, while cleavage activity was lost in the strain with a 26-nt crRNA.

Considering that transcription repression was achieved without Cas3 in previous work[22], we further compared the transcriptional repression activity of truncated crRNAs in the presence and absence of active Cas3. We found *sfgfp* repression was slightly stronger in the presence of Cas3, most notably with crRNAs <20-nt (Supplementary Fig. 5c). These cumulative results suggested that full-length crRNAs directed the cascade complex to target complementary DNA resulting the Cas3 helicase-nuclease recruitment and subsequent cleavage activity, while shorter crRNAs (≤ 26-nt) guide transcriptional repression by the cascade complex. Thus, by controlling crRNA length, the nuclease-based cleavage can be switched to transcriptional repression using the type I-B system from *P. thermoglucosidasius*.

## Characterizing PAM and primed adaptation

Since PAM sequence determines the specificity of crRNA targeting via complementary base pairing, we aligned the 5-nt upstream sequences

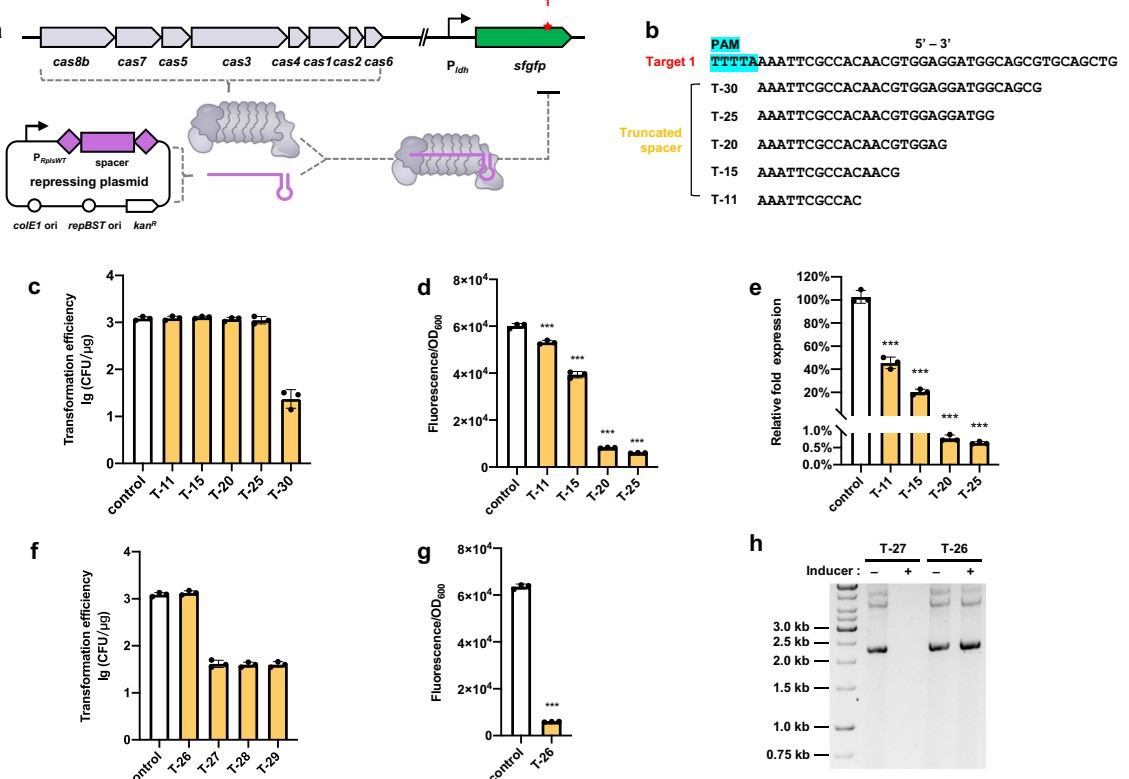

**Fig. 2 | Transcription repression by using shortened crRNA. a** Principle for evaluating transcription repression using sfGFP expression. **b** Design of truncated noncanonical spacers used for generating crRNA. **c** Transformation efficiency of these plasmids expressing the full-length and truncated spacers. Transformation efficiency is determined by the colony recovery rate relative to control plasmid without any spacer targeting the genome. **d** Fluorescence intensity repressed by these crRNAs. The exact *p* values for T-11, T-15, T-20, and T-25 were 0.00072, 2.98e −5, 1.06e−7, and 8.76e−8, respectively. **e** Repression of *sfgfp* gene characterized at transcription level using RT-qPCR. The exact *p* values for T-11, T-15, T-20, and T-25 were 0.00018, 1.88e−5, 5.66e−6, and 5.62e−6, respectively. **f** Transformation efficiency of the plasmids expressing truncated spacers from 26 to 29 nt. **g** Fluorescence intensity repressed by 26-nt crRNA. *P* value = 7.76e−8. **h** Effect of different length crRNA on the persistence of the cognate target plasmid. The test plasmid was purified from equal volumes of cultures of strain T-27 and T-26 grown for 4 h with or without crRNA inducers and resolved by agarose gel electrophoresis. Data are the mean of three biological repeats and are expressed as mean ± SD. Statistical significance is calculated based on two-tailed Student's *t* test (***P < 0.001). Source data are provided as a Source Date file.

of the three protospacers that act as PAMs above to obtain a consensus PAM motif (Fig. 3a). To determine the exact PAM sequence and test whether variants of this sequence could perform the same function, we mutated each nucleotide of the 5-nt putative PAM to saturation, and screened these PAM variants by plasmid interference assays. Based on logo graphs of nucleotide frequency at each position, we examined the effects of the poorly conserved nucleotides at positions −5 and −4 of the putative PAM by generating 16 plasmids with each different nucleotide at the −5 and −4 positions and fixed TTA sequence at positions −3, −2 and −1 (i.e., pNNTTA, N = A, T, C or G) for interference assays. The high similarity in transformation efficiency between these plasmids suggested that the −5 and −4 nucleotides of the putative PAM were independent (Fig. 3b). Next, we tested the influence of all 64 possible tri-nucleotide combination at positions −3, −2, and −1 on interference efficiency (Fig. 3c). We identified TTA as the most effective PAM for target-cleavage-based interference activity. In addition, the PAM sequences of CTA, TCA, TTG, and TTT also displayed interference activity, but the efficiency was significantly lower by two orders of magnitude when compared to TTA (Fig. 3c).

To determine whether the 3-nt TTA PAM was qualified for transcriptional repression in addition to target cleavage, we generated plasmid constructs with either the 3-nt TTA PAM or the 5-nt PAM inserted upstream of the P$_{xylA*}$ promoter (Supplementary Fig. 6) driving *sfgfp* expression (Fig. 3d). We found that this insertion had no influence on fluorescence intensity when transformed into strain Y1 (Fig. 3e). In repression assays testing the effects of the xyla25 crRNA on

*sfgfp* expression, we found no obvious difference in repression activity between the 3-nt TTA or 5-nt PAM (Fig. 3e), indicating that TTA was precise PAM responsible for both transcriptional repression and DNA cleavage. Further, to evaluate the specificity of this PAM sequence for repressing transcription across the whole genome, we performed RNA-seq analysis of strain Y1 harboring the TTA PAM plasmids with or without the xyla25 spacer. This analysis revealed that only *sfgfp* transcripts were decreased in abundance in the presence of the xyla25 crRNA, with no other genes showing significant changes (Fig. 3f). These results supported that the type I-B CRISPR-Cas system with TTA PAM could direct highly specific transcriptional repression without any significant off-target effects beyond random mutation.

To investigate whether the ≤26-nt fully complementary crRNAs elicited primed adaptation (i.e., the acquisition of new CRISPR spacers), we examined the CRISPR arrays in the transformed strains that could have potentially expanded through the addition of new spacers if primed self-adaptation occurred. Although none of the CRISPR arrays were expanded in transformants harboring crRNAs with ≤25-nt complementary sequence, we found that transformants expressing 26-nt complementary sequence indeed showed primed adaptation after 5 days of subculture (Fig. 3g). Sequencing of the newly acquired spacers revealed that they all matched the genome of strain Y1. In addition, we constructed a position frequency matrix based on alignments of the upstream matching sequence, which clearly showed TTA PAM sequence, further supporting TTA as the exact PAM sequence (Supplementary Fig. 7a).

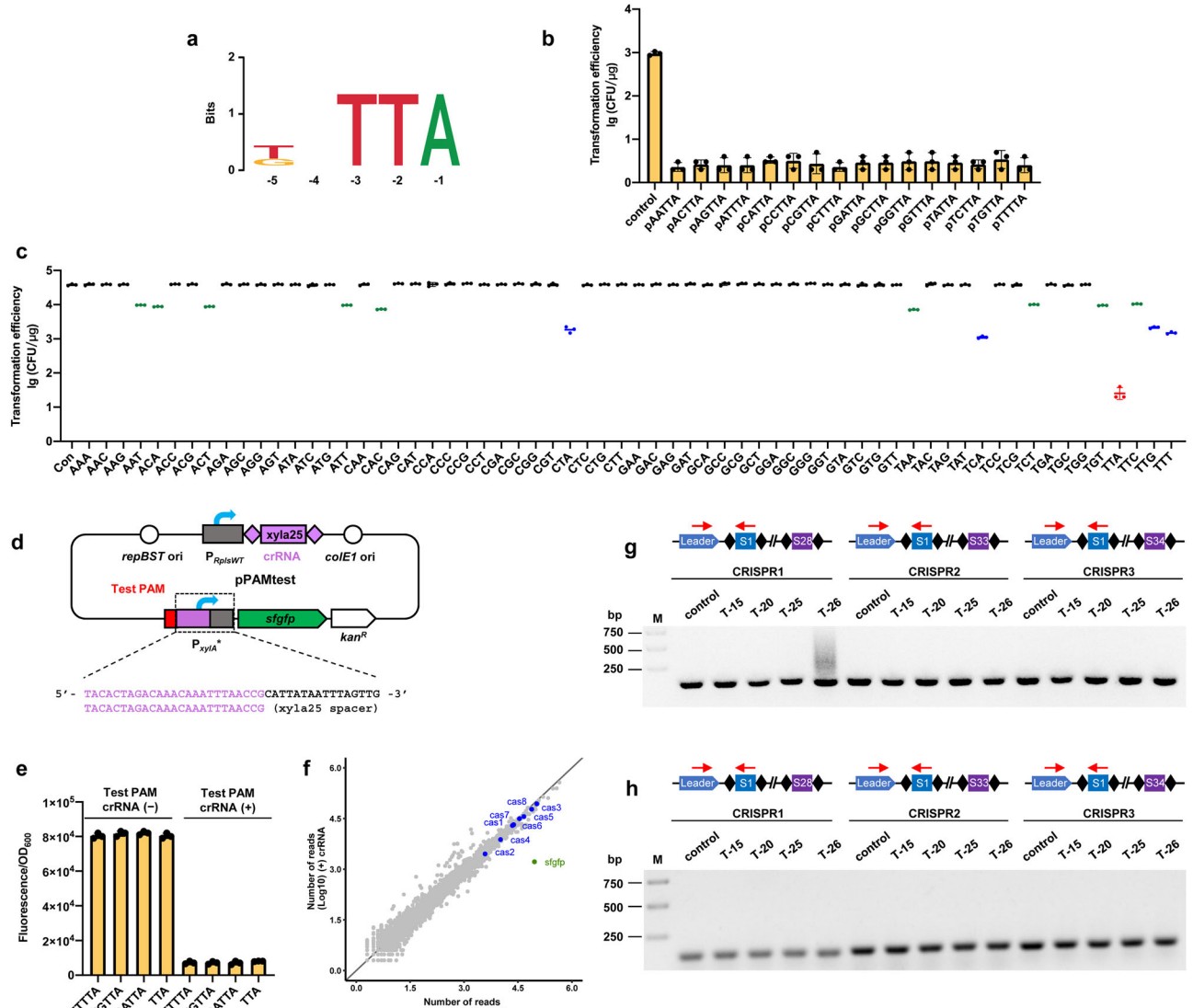

**Fig. 3 | Characterization of this type I-B CRISPR-Cas system. a** Logo showing the putative PAM motif based on three identified protospacer. The sequences were aligned and analyzed with WebLogo (http://weblogo.berkeley.edu/logo.cgi). **b** Effect of mutants at positions −5 and −4 of PAM on interference activity. Interference activity is determined by the colony recovery rate of targeting plasmid transformation. **c** Effect of mutants of the 3-nt PAM on interference activity. **d** Schematic illustration of the developed sfGFP report system for evaluating the specific PAM. The detail of P$_{xylA}$* was shown in Supplementary Fig. S6A, B. **e** Comparison the repression of the 3-nt and 5-nt PAM sequence on sfGFP report system

by measuring fluorescence intensity. **f** Evaluating the specificity of transcription repression on a genome-wide scale based on deep sequencing (RNA-seq). **g** Evaluation of prime adaptation by truncated crRNA. CRISPR expansion at the leader end of CRISPR1, CRISPR2 or CRISPR3 were assayed by PCR. DNA sample from no crRNA expression (control) or truncated crRNA expression (T-26, T-25, T-20, or T-15) strain was used as PCR templates. Lane M, dsDNA size marker. **h** Evaluation of prime adaptation by truncated crRNA. Data are the mean of three biological repeats and are expressed as mean ± SD. Source data are provided as a Source Date file.

We then deleted Cas1-2 and Cas4, which are known to mediate primed adaptation[33], from strain Y2 to generate strain Y3 (Supplementary Fig. 7b). As expected, neither cleavage-based DNA interference nor transcriptional repression of the *sfgfp* target were impaired in the strain Y3 (Supplementary Fig. 7c, d), and the primed adaptation signature disappeared (Fig. 3h). These results led us to conclude that only the effectors Cas3 and Cas5–8 were required for orthogonal DNA cleavage and transcriptional repression activity, suggesting that we could design a transferrable tool for heterologous genome engineering using these essential proteins.

### Orthogonal tool development in both thermophile and mesophile

We next examined whether the targeting location of crRNAs affected their transcriptional repression activity. First, comparison of *sfgfp*

fluorescence by targeting at the same site on template versus non-template strands (i.e., crRNA2 and crRNA3 target sites, Supplementary Fig. 8a, b) showed stronger suppression of *sfgfp* on the template strand (Supplementary Fig. 8c). Subsequent targeting of different positions on the template strand (i.e., spacers 1, 2, or 4) (Supplementary Fig. 8a) showed no significant differences in repression activity of truncated crRNAs targeting upstream, midstream, or downstream sites (Supplementary Fig. 8c).

We then tested whether this system could be applied to induce gene knockout using the amylase gene (*BCV53_04180*) in strain Y1 for a proof-of-concept example. We constructed an editing plasmid with crRNA and homologous arms (Fig. 4a) and introduced it into strain Y1, which yielded few clones compared to the control vector (Fig. 4b). We hypothesized that constitutively expressed crRNA resulted in excessive Cas-mediated cleavage activity that could not be repaired by less

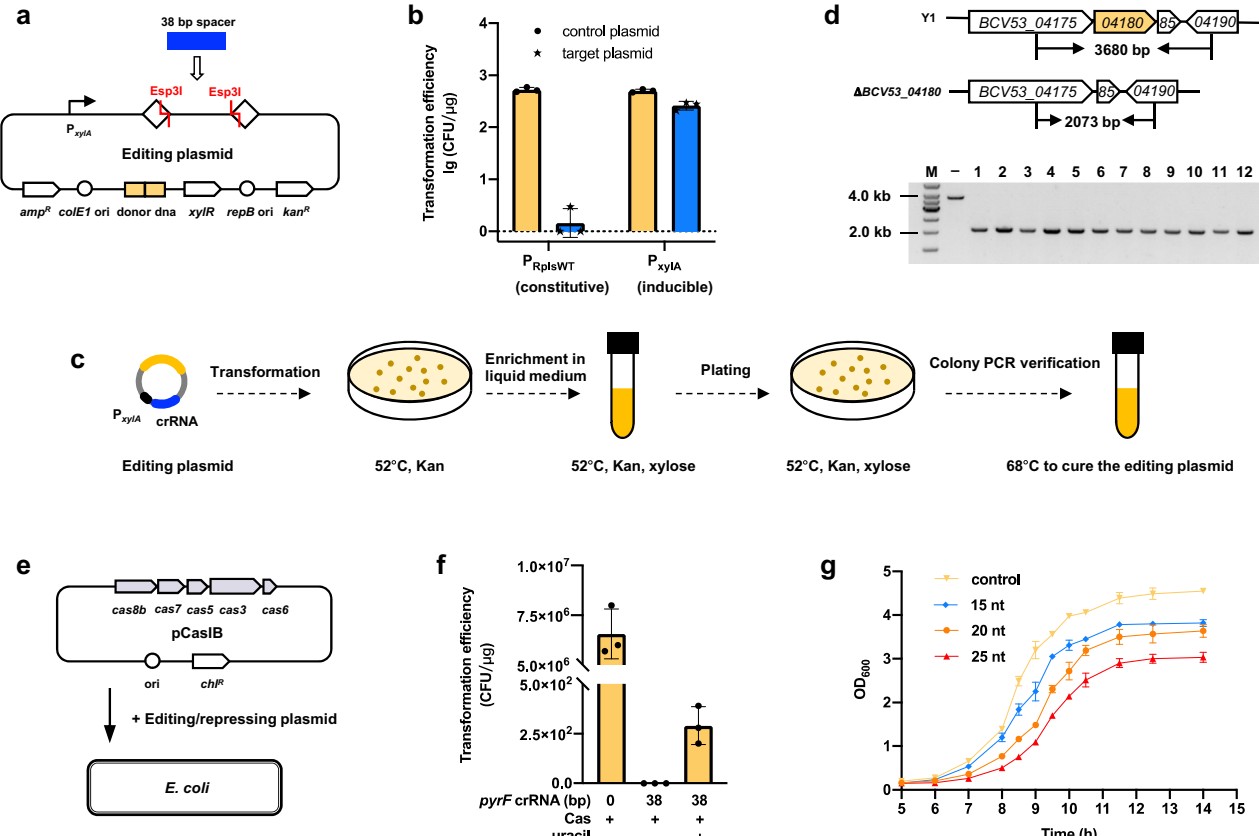

**Fig. 4 | Orthogonal tool development in thermophilic *P. thermoglucosidasius* and mesophilic *E. coli*. a** Schematic of the editing plasmid. This plasmid equipped Esp3I cleavage sites for the insertion of crRNA. ColE1 ori: replication origin used in *E. coli*, *amp*^R: ampicillin resistance gene, R: repeat, *lacZ* used for blue-white selection of the spacer insertion, repB ori: replication origin used in *Geobacillus* or *Parageobacillus* genus, *kan*^R: kanamycin resistance gene. **b** Transformation efficiency of targeting amylase gene for allelic recombination based on deletion. $P_{RplsWT}$ and $P_{xylA}$ indicate the constitutive and inducible promoter driving crRNA, respectively.

**c** The workflow of deletion in *P. thermoglucosidasius*. **d** Confirmation of amylase gene deletion by PCR. **e** Schematic of the transferrable plasmid pCasIB harboring the type I-B Cas effectors. The plasmid pCasIB and editing/repressing plasmid together were used to editing or repressing target gene in *E. coli*. **f** Deleting *pyrF* gene in *E. coli* using the transferrable tool. **g** Growth rate influenced by the transcription repression of *pyrF* gene. Data are the mean of three biological repeats and are expressed as mean ± SD. Source data are provided as a Source Date file.

---

efficient DNA recombination mechanisms[34]. To reduce crRNA expression, we used a well-characterized[35] inducible promoter, $P_{xylA}$ (Fig. 4a and Supplementary Table 1), which resulted in markedly higher transformation efficiency without inducing crRNA after transformation (Fig. 4b). The sample knockout steps are shown in Fig. 4c and the successful deletion of the amylase gene without need for marker selection was confirmed (Fig. 4d). These results provided strong evidence that this type I-B CRISPR-Cas system could be used as genome-editing tool in *P. thermoglucosidasius*.

We next sought to determine whether this tool was similarly effective in other (i.e., mesophilic) microbial hosts. Using *E. coli* W3110 as an example due to it wide use in chemical production applications[36], we introduced the type I-B Cas effectors Cas3 and Cas5–8 on a transferable plasmid (pCasIB, Fig. 4e) compatible with the editing/repression plasmid (Figs. 2a and 4a). To test the activity of this modular format, we used a crRNA targeting the *mals* gene in the *E. coli* W3110 genome. We observed that co-expression of pCasIB with this 38-nt crRNA resulted in potent interference of *mals* (Supplementary Fig. 9a). The gene *pyrF*, encoding orotidine-5′-phosphate decarboxylase in the pyrimidine biosynthetic pathway, a marker for pyrimidine auxotrophy[37], was selected for deletion or repression. After introducing pCasIB and an editing plasmid carrying a 38-nt crRNA targeting *pyrF* and homologous arms into *E. coli* W3110, clones were recovered on plates with 250 μmol uracil, but not on plates lacking uracil, which indicated that PyrF function was impaired in

transformants (Fig. 4f). After confirming *pyrF* deletion (Supplementary Fig. 9b), we delivered repression plasmids with 15-nt, 20-nt, or 25-nt crRNAs targeting *pyrF*, individually, into *E. coli* W3110 containing pCasIB to test the repression effects on *pyrF* gene. Following selection on uracil+ medium, colonies were cultured in uracil− medium, which led to a gradual decrease in biomass formation that aligned with increasing crRNA length from 15- to 25-nt (Fig. 4g). These results indicated that crRNA length could be used to tune the transcription levels of target genes in *E. coli* (Supplementary Fig. 9c). These results further demonstrated the capacity of the thermostable type I-B system for efficient genome editing with concurrent gene repression in mesophilic bacterial species.

**Improving transformation efficiency using this orthogonal tool**
In general, tools developed by CRISPR-Cas systems exponentially increase the speed of genetic manipulation if the microorganisms hold the capability of efficient transformation. However, *P. thermoglucosidasius* is relatively recalcitrant to the incorporation of foreign DNA (i.e., low-transformation efficiency), which presents another major obstacle to its use as a thermophilic microbial chassis. We therefore sought to identify the polygenic factors responsible for precluding the incorporation of foreign DNA using the orthogonal I-B CRISPR-Cas system and to engineer a *P. thermoglucosidasius* with high transformation efficiency (see Fig. 5a for workflow). First, iterative deletions are introduced to remove the putative endogenous defense

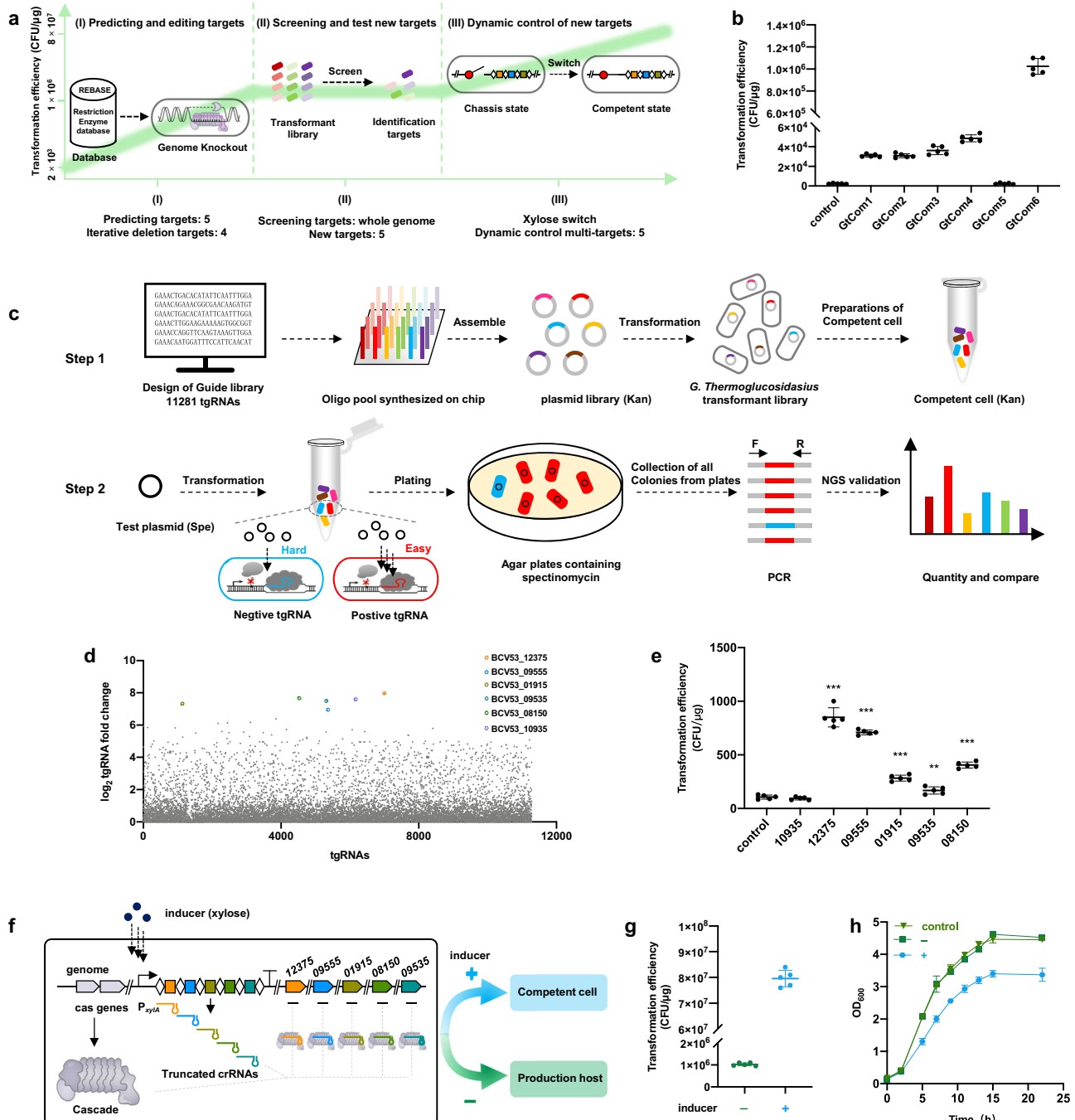

**Fig. 5 | Workflow for systematic improvement of transformation efficiency.**
**a** Workflow for improving transformation efficiency. **b** Transformation efficiency of strain GtCom1–GtCom6. Control: WT strain. GtCom1 (Δ*BCV53_09985*), GtCom2 (Δ*BCV53_05690*), GtCom3 (Δ*BCV53_12685*), GtCom4 (Δ*BCV53_09960*), GtCom5 (Δ*BCV53_8635*), GtCom6 (Δ*BCV53_09985*, Δ*BCV53_05690*, Δ*BCV53_12685*, and Δ*BCV53_09960*), n = 5 biologically independent samples. **c** Schematic for screening the new targets contributing to transcription efficiency. **d** Abundance changes of tgRNAs assayed by deep sequencing. **e** Testing the transformation efficiency of strain GtCom6 by re-transform selected positive tgRNAs, *n* = 5 biologically independent samples. The exact *P* values for *BCV53_10935*, *BCV53_12375*, *BCV53_09555*,

*BCV53_01915*, *BCV53_09535*, and *BCV53_08150* were 0.25, 8.69e−8, 6.05e−11, 2.62e−6, 0.0076, and 5.09e−8, respectively. **f** Schematic of xylose-induced CRISPR array containing multiple tgRNAs. Strain status switch between competent cell and production host by inducer. The symbol '+' indicates the medium with inducer, and '−' indicates without. **g** Transformation efficiency of strain GtCom7 with or without inducer, *n* = 5 biologically independent samples. **h** Growth rate of strain GtCom7 with or without inducer. Data are the mean of three biological repeats and are expressed as mean ± SD. Statistical significance is calculated based on two-tailed Student's *t* test (**P < 0.01; ***P < 0.001). Source data are provided as a Source Date file.

systems. Then, a screening strategy was designed to identify the unknown targets that contribute to blocking genetic transformation across the genome. Finally, we integrated these targets to produce dynamically controlled competent cells with high transformation efficiency when induced. The orthogonal I-B CRISPR-Cas tool was an integral component of the entire process.

(I)  Iterative deletion of putative natural defense targets. Native restriction-modification (R-M) systems cleave specific sites in double-stranded exogenous DNA. In strain NCIMB 11955, five restriction enzyme genes (*BCV53_09985*, *BCV53_05690*, *BCV53_12685*, *BCV53_09960*, and *BCV53_08635*) were predicted by the Restriction Enzyme database (REBASE)[38]. To test the effects

of these genes on transformation efficiency, each of the five genes were individually deleted from strain Y1 to obtain strains GtCom1–GtCom5, respectively (Supplementary Fig. 10a). Evaluation of transformation efficiency using plasmid pUCG3.8 in these strains showed that deletion of each of these candidates, except *BCV53_08635*, could increase transformation efficiency compared to that in control strain Y1 (Fig. 5b). Since growth curves of strains GtCom1–GtCom4 indicated that the loss of these genes did not affect *P. thermoglucosidasius* growth (Supplementary Fig. 10b), successive deletion of *BCV53_09985*, *BCV53_05690*, *BCV53_12685*, and *BCV53_09960* in strain Y1 to generate strain GtCom6 resulted in ~500-fold higher transformation efficiency than that of strain Y1 ($\sim 1 \times 10^6$ CFU/μg) (Fig. 5b). It is worth noting that these genetic manipulations required less than one month using this tool.

(II) Identification of the previous unknown targets. Although this transformation efficiency was markedly higher than that of wild-type NCIMB 11955, it remained much lower than that of *E. coli* or *Bacillus subtilis*. We then developed a screen strategy for other genes throughout the genome involved in hindering genetic transformation using this orthogonal tool (see experimental scheme in Fig. 5c). First, we constructed a genome-scale CRISPR interference (CRISPRi) library with 11,281 truncated guide RNAs (tgRNAs) (Supplementary Data 2) targeting 3821 of 3844 total genes (99.94%) in *P. thermoglucosidasius* genome (Fig. 5c and Supplementary Fig. 10c). To ensure the transcriptional repression of each gene, three tgRNAs were designed for 95.9% of the genes, targeting upstream, midstream and downstream of template strand (Supplementary Fig. 10c). The tgRNA pool was assembled into a repression plasmid to produce a plasmid library (Fig. 5c and Supplementary Fig. 10d), and evaluated coverage of the plasmid library through NGS (Supplementary Table 2), indicating that tgRNAs in plasmid library covered almost 100%. The plasmid library was then transformed into strain GtCom6 to generate a collection of strains that theoretically contained at least one strain expressing each individual tgRNA. If a tgRNA led to repression of a given gene that blocked transformation, the strain harboring that tgRNA should more readily accept the test plasmid than strains with tgRNAs unrelated to transformation efficiency. Strains with higher transformation rates of the test plasmid due to the tgRNA should accordingly grow more clones on selective media. Deep sequencing of the transformed clones identified several tgRNAs with higher relative abundance (Fig. 5c). As expected, we detected a set of high abundance tgRNAs targeting several genes, including *BCV53_12375*, *BCV53_09555*, *BCV53_01915*, *BCV53_09535*, *BCV53_08150*, and *BCV53_10935* (Fig. 5d and Supplementary Table 3). To confirm that these targets negatively affected transformation efficiency, these positive tgRNAs were re-transformed into strain GtCom6, which substantially increased transformation efficiency over that of the control, except for the tgRNA targeting *BCV53_10935* (Fig. 5e), indicating that the five newly identified targets indeed influence transformation efficiency. To our knowledge, this is the first time targets involved in transformation efficiency were screened across the whole genome in bacteria.

(III) Dynamic control of transformation competency. Functional assays suggested that direct deletion the identified targets could potentially affect the performance of the strain, such as growth rate (Supplementary Table 3). For instance, knockout of *BCV53_08150*, encoding gluconate 5-dehydrogenase, resulted in decreased growth rate (Supplementary Fig. 10e, f). To address this issue, we used the orthogonal tool to temporarily and simultaneously inhibit the expression of these targets specifically during transformation. We constructed a set of tgRNAs targeting the five candidate's transformation-related genes driven by a xylose-induced promoter, and introduced these constructs into the genome of GtCom6 to create strain GtCom7 (Fig. 5f and Supplementary Fig. 10g, h). Following induction of tgRNA expression, transformation efficiency of strain GtCom7 reached $8 \times 10^7$ CFU/μg DNA, about two orders of magnitude higher than GtCom6 (Fig. 5g). Growth curves for GtCom7 with xylose induction showed that growth was considerably slower after induction relative to that of GtCom6, but highly similar to that GtCom6 without xylose induction (Fig. 5h). These results indicated that xylose could induce a switch between the production host and the competent cell in GtCom7 cells (Fig. 5f). Evaluation of tgRNA stability in the GtCom7 genome after five passages in flask cultures indicated no recombination events or other deleterious mutations occurred in this design (Supplementary Fig. 10i). These results demonstrated the use of our orthogonal tool for a powerful genetic screen and a following multiplexed engineering to achieve dynamic control of transformation competency in a recalcitrant, thermophilic, non-model species. Through these strategies, we generated the highest transformation efficiency *P. thermoglucosidasius* strain reported to date.

## Construction of thermophilic riboflavin cell factory by genome-scale identification of and combinatorial manipulation of multiple targets

Having addressed the bottleneck of genetic manipulation and high-efficiency transformation in *P. thermoglucosidasius*, we aimed at rapidly de novo constructing a thermophilic riboflavin cell factory using *P. thermoglucosidasius* GtCom7 strain (Fig. 6a). We first introduce a copy of riboflavin biosynthetic gene cluster, high-performance liquid chromatography (HPLC) analysis showed significantly increased riboflavin production in fermentations (Fig. 6b and Supplementary Fig. 11a, b). The developed type I-B CRISPR tool was employed to rapidly integrate the previously known targets (i.e., deleting *purA*, *purR*, *ccpN*, and *ldh* gene and mutating *ribC* gene)[32], resulting in strain Gt-06 with the titer of 421 mg/L (Fig. 6b and Supplementary Fig. 11c). To avoid feedback inhibitions of purine pathway for the biosynthesis of the precursor guanosine 5′-triphosphate (GTP), we tested three mutants of PurF that have confirmed to release feedback inhibition in *B. subtilis*[39], indicating that D292V in the Gt-06 strain (i.e., Gt-07 strain, Supplementary Fig. 11d, e) contribute to a higher riboflavin titer compared to the other residue mutations (Supplementary Fig. 11f). Further, we replaced the native promoter of *pur* operon with a constitutive promoter $P_{YceD}$ (Supplementary Table 4) to decouple the riboswitch control, generating the Gt-08 strain with increasing riboflavin titer to 521 mg/L in shake flask fermentation (Fig. 6b and Supplementary Fig. 11c, g).

To identify more unknown targets, the above genome-scale CRISPRi library was used to rapidly screening based on the observable colorimetric phenotype of riboflavin (Fig. 6a, c). In total, 16 colonies displaying a much yellower color were selected for further fermentation verification. Twelve out of 16 colonies exhibited enhancements in riboflavin titer relative to the control Gt-08 strain, with 10 displayed significant increases ($P < 0.001$, Supplementary Fig. 12a and Table 5). Subsequently, a repressing plasmid containing a CRISPR array with five truncated spacers that simultaneously targeted the five most effective targets was transformed into the Gt-08 strain, generating strain Gt-09 (Fig. 6e and Supplementary Fig. 12b). The riboflavin titer of Gt-09 showed 38% increase compared to that of Gt-08, reaching a maximum titer of 719 mg/L (Supplementary Fig. 12c). Finally, in a 5-L bioreactor, the performance of the engineered Gt-09 strain produced riboflavin with a final titer of 5.03 g/L within 28 h (Fig. 6f), which represents the highest riboflavin titer achieved in thermophilic *P. thermoglucosidasius*. Notably, the fermentation was performed under the conditions that the bioreactor and the carbon substrates were not subjected to sterilization, and we did not observe any bacterial

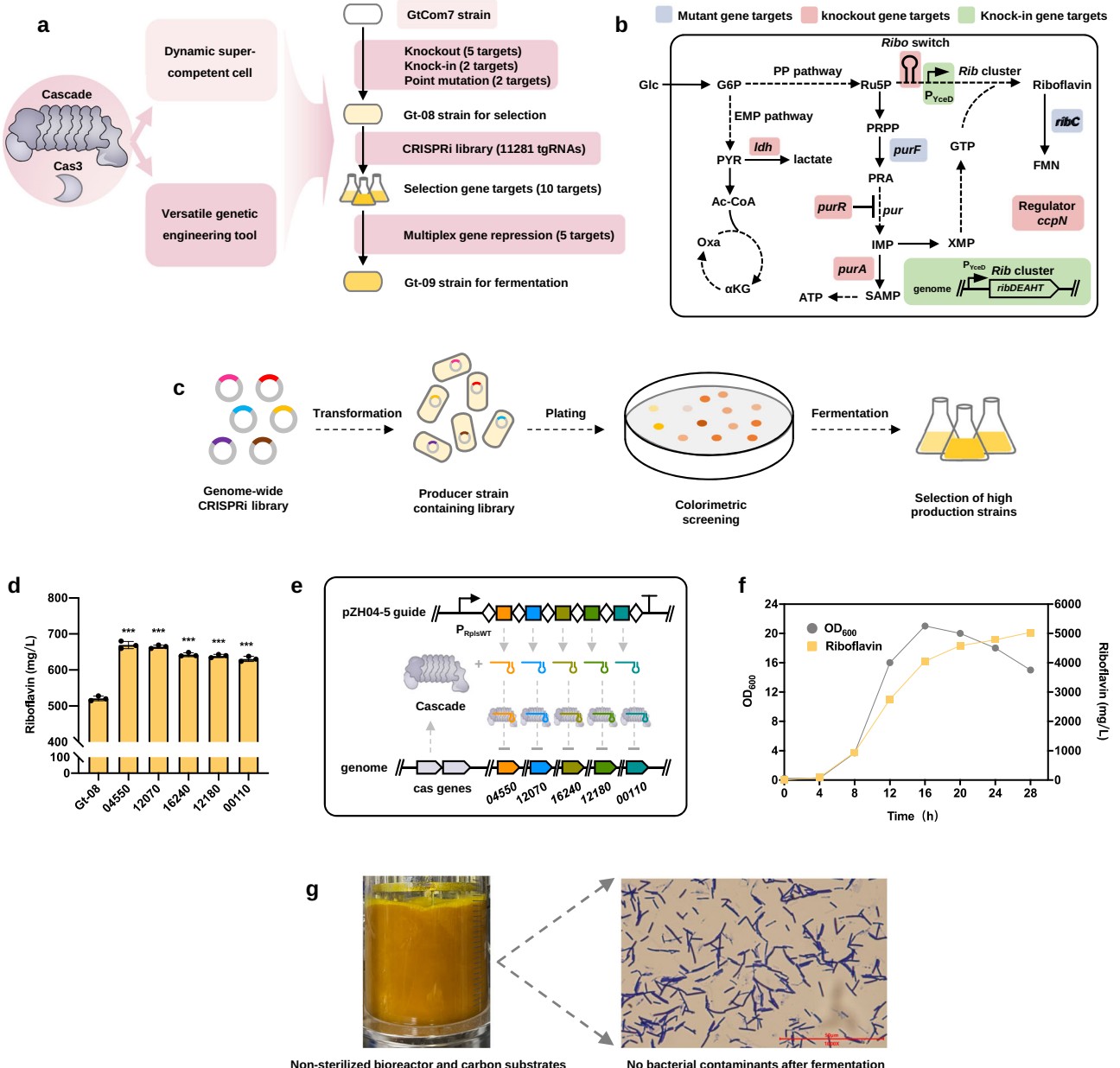

**Fig. 6 | Workflow for systematic improvement of riboflavin production.**
**a** Workflow for improving riboflavin production. **b** Metabolic pathways for the biosynthesis of riboflavin. Abbreviations in Supplementary Note 1. **c** Schematic for colorimetric screening the new targets contributing to riboflavin production. **d** Testing the riboflavin production of strain Gt-08 by re-transform selected positive tgRNAs. The exact $p$ values for *BCV53_04550*, *BCV53_12070*, *BCV53_16240*, *BCV53_12180*, and *BCV53_00110* were 2.83e−5, 6.23e−6, 1.62e−5, 1.39e−5, and 3.86e

−5, respectively. **e** Schematic of CRISPR array containing multiple tgRNAs expressed in the plasmid. **f** Fed-batch fermentation profile of the final riboflavin strain Gt-09. **g** Microscopic examination for identifying no bacterial contaminants. Data are the mean of three biological repeats and are expressed as mean ± SD. Statistical significance was calculated based on two-tailed Student's $t$ test (***$P < 0.001$). Source data are provided as a Source Date file.

contaminants by both microscopic evaluation and plate culture (Fig. 6g and Supplementary Fig. 12d) during fermentation, further underscoring the potential of thermophilic *P. thermoglucosidasius* as a promising chassis organism.

## Discussion
The potential for high-temperature biomanufacturing has recently gained attention due to the discovery and application of several high-efficiency, thermophilic bacterial strains[15,40,41]. High temperatures similar to those used in chemical refineries enable faster fermentation, and thus shorter production cycles with reduced cooling costs and risk of contamination. However, a lack of tools for genetic modification

and low-transformation efficiency hinders the advanced exploration and development of thermophilic chassis strains, as well as our fundamental understanding of their physiology and metabolism. In this study, we address these two key bottlenecks by developing a tool for orthogonal genetic manipulation and a strategy for dynamically inducible high-efficiency transformation, which together provide a straightforward and powerful platform for thermophile research and engineering (Fig. 7).

While CRISPR technology has revolutionized gene manipulation, thermostable Cas9 proteins, such as ThermoCas9[42], GeoCas9[43], and IgnaviCas9[44] have inherent inconveniences that prevent their use in engineering *P. thermoglucosidasius*. This inherent limitation is likely

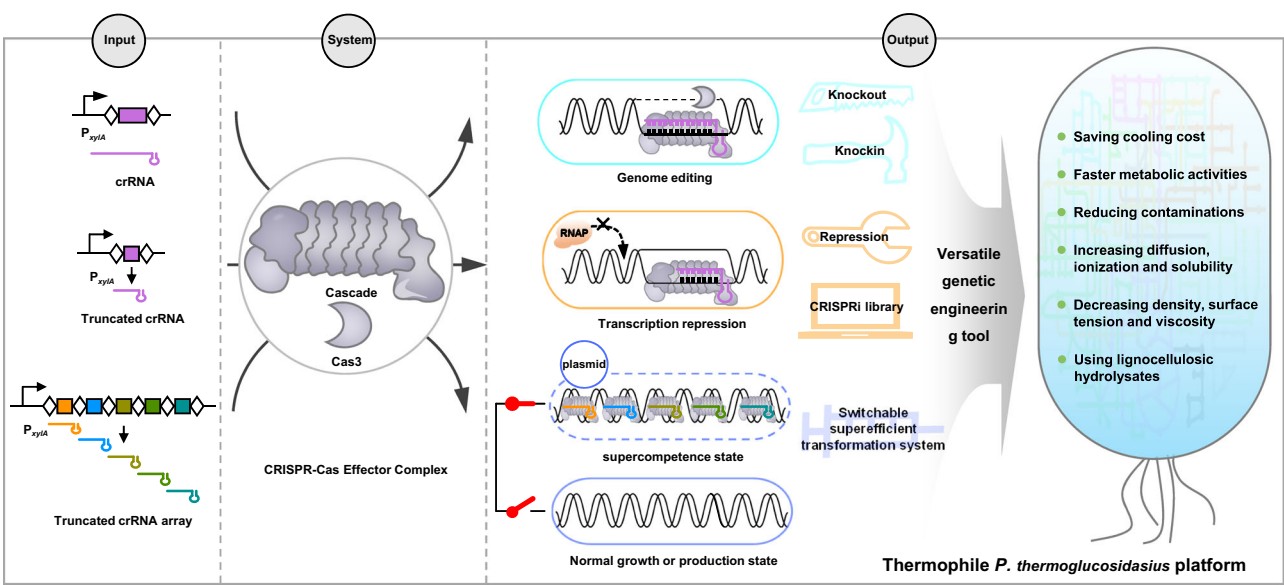

**Fig. 7 |** A versatile genetic engineering tool contributing to building thermophilic factories.

due to the promiscuous recognition of multiple PAM sequences by these Cas9s (i.e., NNNNCNAA or NNNNCMCA for ThermoCas9, NNNNCRAA for GeoCas9, and NRRNAT for IgnaviCas9), as well as the relatively great PAM length[42–47]. Given that PAM sequences play a crucial role in distinguishing self vs. non-self during the interference stage, the relatively longer and promiscuous PAM sequences associated with these thermostable Cas9 proteins may rasie concerns about potential toxicity and off-target effects[48,49]. In contrast with these relatively non-specific PAM targets, the endogenous type I-B CRISPR-Cas system identified here recognizes a specific TTA PAM (Fig. 3a) that results in no detectable toxicity or off-target effects in *P. thermoglucosidasius* (Fig. 3f). This 3-nt PAM also enables the easy design of complementary crRNAs (e.g., we designed specific, complementary crRNAs for 99.94% of genes in the *P. thermoglucosidasius* genome; Supplementary Fig. 10c).

More importantly, our results demonstrate that control crRNA length alone enables switching between genome editing and transcriptional repression activities, and does not require deleting or inactivating Cas3 nuclease in *P. thermoglucosidasius*. This finding aligns well with previous work that showed natural or engineered transcriptional repression by a type I-B system in *H. hispanica* that could be switched by altering crRNA length[23,24], although a 24-nt spacer is sufficient to induce the functional switch between interference and repression in *H. hispanica*, while 26-nt spacer is required in *P. thermoglucosidasius*. We speculated that the shorter crRNAs could still guide the cascade proteins to target DNA, but might not activate nuclease activity, which may be a conserved interaction among type I-B Cas proteins, crRNAs, and DNA. Konstantin et al. found that shortened crRNAs result in formation of Cas complexes with fewer subunits than those formed with full-length crRNAs in the type I-E CRISPR-Cas system in *E. coli*[50], implying crRNA length could alter the stoichiometry of cascade protein complex interactions leading to a functional switch, although no evidence showed that shortened crRNAs could guide gene repression in that system[50]. Future work will more closely examine the relationship between crRNA length and prime adaptation, interference, or transcriptional repression activity in type I CRISPR-Cas systems.

In addition to gene manipulation, efficient transformation is essential for investigation and redesign of the genetic programs underlying complex phenotypes[51]. Bacteria have evolved diverse defense mechanisms against exogenous DNA, including native restriction-modification enzymes and CRISPR-Cas systems. However, engineering only these restriction endonuclease targets results in

persistently low-transformation efficiency, compared to *E. coli* and *B. subtilis*[52], that limits further application in non-model strains. Using the orthogonal tool developed here, we established a systematic workflow for improving transformation efficiency (Fig. 5a) in any given strain by screening for target genes throughout the genome that contribute to transformation efficiency. This study also reports, to the best of our knowledge, the first genome-scale CRISPRi library using tgRNAs to guide the type I-B CRISPR-Cas orthogonal tool to screen potential genotype–phenotype interactions (Fig. 5d). Among the five candidate targets, two targets were homologs previously described to play a role in transformation in other organisms. First, *BCV53_12375* encodes mannose-1-phosphate guanylyl transferase and reportedly catalyzes mannose-1-phosphate conversion to GDP-mannose in lipopoly-saccharide (LPS) biosynthesis[53]. The other is *BCV53_09555*, encoding the DNA mismatch repair protein, MutS, the downregulation of which reportedly leads to higher acquisition of foreign DNA by natural transformation in *Pseudomonas stutzeri*[54]. The remaining three targets, *BCV53_01915*, *BCV53_09535*, and *BCV53_08150*, have not been previously documented to participate in regulating the fate of foreign DNA. These findings indicate that genome-scale phenotypic screening is a powerful approach to identifying genes involved in transformation efficiency, and supporting the further application of this orthogonal tool in increasing the competence of other non-model, recalcitrant strains.

It also warrants mention that we established dynamically controlled, transformation-competent *P. thermoglucosidasius* cells by leveraging the multi-gene repression capability of this orthogonal tool. This strategy prevented strain damage or lethality that inevitably occur when implementing successive, multiplex gene knockout of the identified targets[55,56]. To perform genetic transformation, the inducer is added to the medium to turn on the competent state. In medium without the inducer, xylose, the cells remain healthy and metabolically active. Thus, the use of an inducible promoter to drive the repression of multiple targets results in a dynamic switch that preserves cellular integrity.

Empowered by efficient genetic manipulation and transformation efficiency have been developed in *P. thermoglucosidasius*, we were able to de novo construct a riboflavin overproduction cell factory within two months. The identification of unkown targets that involve in riboflavin production again demonstrated the effectiveness of CRISPRi library in genome-scale inquiry of "genotype–phenotype" correlation (Fig. 6c). In addition, this type I-B CRISPR-Cas system provides a conveniently approach to simultaneously modify multiple gene targets, as shown in both the dynamic competent cell and the engineered

riboflavin production strain. Furthermore, considering the level of transcriptional repression could be adjusted by the spacer length in the CRISPR array (Fig. 2d), we anticipated that this tool could also qualified for multiplexed fine-tuning for further strain improvement.

Overall, we extended the capabilities of the thermostable type I-B CRISPR-Cas system that enabled diverse genetic engineering efficiently in *P. thermoglucosidasius*. This significant advancement addressed two major obstacles—high-efficient genetic manipulation and transformation—that hindered the broader understanding and engineering of this species. This work expands the available genetic toolkit and paves the way for engineering *P. thermoglucosidasius* as promising thermophilic cell factories.

## Methods

### Bacterial strains and growth conditions

Bacterial strains used in this work are listed in Supplementary Data 3. *P. thermoglucosidasius* NCIMB 11955 were purchased from China General Microbiological Culture Collection Center (CGMCC). *Escherichia coli* JM109 and W3110 were routinely cultivated at 37 °C in Luria-Bertani (LB) medium. *P. thermoglucosidasius* NCIMB 11955 was growth in mLB medium (0.59 mM MgSO$_4$.7H$_2$O, 0.91 mM CaCl$_2$.2H$_2$O and 0.04 mM FeSO$_4$.7H$_2$O were added to LB medium) or on the TSA (Oxoid™, CM0131) agar plate. For fermentation of *P. thermoglucosidasius*, riboflavin fermentation medium (detailed in Supplementary Method 1) was used in 250 ml flasks at 250 rpm and 60 °C. When required, antibiotics were added to the growth media at the following concentrations: 20 mg/L kanamycin (Km), 25 mg/L chloromycetin (Cm) or 100 mg/L of ampicillin for *E. coli* and 12.5 mg/L Km or 10 mg/L spectinomycin (Sp)[57] for *P. thermoglucosidasius*.

### Plasmids construction and transformation

The primers used in this study were listed in Supplementary Data 4. The method is detailed in Supplementary Method 2. *E. coli* strains were genetically manipulated according to the standard protocols[58]. Transformation *P. thermoglucosidasius* strains was achieved through electorporation[16]. In short, electroporation was performed using a solution comprising 0.5 M mannitol, 0.5 M sorbitol, and 10% (V/V) glycerol for this process. For the electroporation, 60 μL of electro-competent cells were mixed with approximately 500 ng of plasmid DNA. Following the transformation, the cells were allowed to recover by incubating them in pre-warmed TGP medium at a temperature of 52 °C for 1–4 h with continuous shaking. To select the transformed cells, The cell suspension was plated onto a TGP agar medium containing 12.5 mg/L kanamycin at 52 °C. After about 24-h incubation, the transformants were recovered on the plate.

### Plasmid or genome interference assays

For the plasmid or genome interference assay, *P. thermoglucosidasius* NCIMB 11955, *E. coli* W3110, or their derived strain was transformed with the target plasmid. The transformants were screened on TSA agar plates for *P. thermoglucosidasius* and their derivants, and LB agar plates for *E. coli* W3110. For each target plasmid, at least three independent replicates were performed.

### RNA extraction and transcription-level assay

Cells were harvested in late-exponential phase and immediately transferred into liquid nitrogen to block the metabolism. Total RNA was isolated using RNAprep Pure Cell/Bacteria Kit (Tiangen Biotech, Beijing, China) according to the manufacturer's recommendation. RNA quantification was carried out by measuring absorbance at 260/280 nm using a NanoDrop ND-2000 spectrophotometer (Thermo Scientific, Wilmington, DE, USA).

For both reverse transcription PCR (RT-PCR) assays and real-time quantitative PCR (RT-qPCR) experiments, the first-strand synthesis of cDNA was performed using 1 μg of total RNA with a PrimeScript™ RT

Reagent Kit with gDNA Eraser (TaKaRa, Japan) following the manufacturer's instructions.

The RT-qPCR procedures were performed as previously described[59]. The experiments were performed using PowerUp™ SYBR™ green master mix (ABI; USA). Results were collected and analyzed using the supporting 7500 software. The expression level of interested genes was calculated using the housekeeping gene *gap* as reference gene[59]. Technical triplicates of three biological repeats were performed per condition.

### RNA-seq

Total RNA was extracted using TRIzol® Reagent (Invitrogen) and genomic DNA was removed using DNase I (TaKara) and high-quality RNA sample (OD260/280 = 1.8–2.0, OD260/230 ≥ 2.0, RIN ≥ 6.5, 28 S:18 S ≥ 1.0, ≥100 ng/μL, ≥2 μg) was used to construct sequencing library.

RNA-seq transcriptome library was prepared for RNA sequencing using TruSeq™ RNA sample preparation Kit from Illumina (San Diego, CA) using 2 μg of total RNA. Libraries were size selected for cDNA target fragments of 200 bp on 2% Low Range Ultra Agarose followed by PCR amplified using Phusion DNA polymerase (NEB) for 15 PCR cycles. After quantified by TBS380, paired-end RNA-seq sequencing library was sequenced with the Illumina HiSeq×TEN (2 × 150 bp read length). Then processing of original images to sequences, base-calling, and quality value calculations were performed using the Illumina GA Pipeline (version 1.6). The data generated from the Illumina platform were used for bioinformatics analysis. All of the analyses were performed using the free online platform of Majorbio Cloud Platform (www.majorbio.com) from Shanghai Majorbio Bio-pharm Technology Co., Ltd.

### Genome editing

To construct strain *P. thermoglucosidasius* 11955Y1 and 11955Y2, the plasmid pUB-IIIBKO and pUB-GFPKI were iteratively introduced into *P. thermoglucosidasius* NCIMB 11955, and the mutant was obtained as our previously described. The other construction of genome editing in *P. thermoglucosidasius* NCIMB 11955 or *E. coli* W3110 were completed by CRISPR type I-B system. Briefly, for *P. thermoglucosidasius* NCIMB 11955 genome editing, the strain transformed with editing plasmid was firstly cultured in TSA agar plate supplemented with 12.5 mg/L Km at 52 °C. Then a single clone from TSA agar plate was cultured in mLB medium supplemented with 25 mM xylose and Km at 52 °C with 250 rpm for 12–24 h. Cultures were serially diluted and plated on TSA agar plate containing 12.5 mg/L Km at 52 °C for 12 h. Clones were verified by colony PCR and Sanger sequencing. To cure the editing plasmid in strain *P. thermoglucosidasius* NCIMB 11955, a single edited clone, including the editing plasmid, was cultured in mLB medium overnight at 68 °C without Km and then diluted and spread on TAS agar plate. The colonies that grew on TSA plates were randomly picked and screened on TSA plates carrying Km. The colonies that were sensitive to Km were free of editing plasmid. For *E. coli* W3110 genome editing, the *E. coli* W3110 electroporation competent cells mixed with editing plasmid and pCasIB was electroporated. Then the electroporation mixture was immediately suspended in 1 ml LB medium including 25 mM xylose. Cells were recovered by culturing at 30 °C for 2 h before spreading on LB plate supplemented with Km, Cm, ampicillin, and 25 mM xylose. Individual colonies were randomly picked from the plate incubated overnight at 37 °C and verified by colony PCR and Sanger sequencing.

### Spacer acquisition assay and protospacer analysis

To monitor spacer acquisition from genomic target by PCR, repression plasmid expressing a series of truncated spacer (i.e., 26 bp spacer, 25 bp spacer, 20 bp spacer, 15 bp spacer, and 11 bp spacer) targeting genomic *sfgfp* gene was transformed into strain 11955Y2. The strain 11955Y2 including expressing spacer were cultured in mLB medium at 60 °C with 250 rpm. Cultured to exponential stage, subsequently transferred

(1:100) into mLB medium and then cultured until stationary stage, the stationary cultures were subjected to PCR analysis as described previously. The protospacer was analyzed as described previously[60].

## Fluorescence measurement

After cultivation to the late-exponential phase, fluorescence intensity and $OD_{600}$ of the cell culture were simultaneously measured using an EnSpire™ Multimode Plate Reader (PerkinElmerInc., USA). The excitation and emission wavelengths of sfGFP were 488 nm and 510 nm, respectively.

## Measurement of biomass and riboflavin

Cell growth was monitored by measuring optical density at 600 nm ($OD_{600}$) with an EnSpire™ Multimode Plate Reader (PerkinElmer Inc., USA). Titers of riboflavin of *P. thermoglucosidasius* strains underwent High-Performance Liquid Chromatography (HPLC) analysis using a Shimadzu Prominence HPLC system equipped with a dual λ UV detector and a YMC polymer C18 column measuring 4.6 × 250 mm. The separation process was carried out under the following conditions: a mobile phase composition consisting of 60% $H_2O$, 10% methanol, 20% acetonitrile, and 10% phosphoric acid (2 mM), maintained at a constant flow rate of 1 ml/min. The concentration of riboflavin was determined by measuring the corresponding peak areas detected at 370 nm[61].

## Producing high-efficiency electrotransformation-competent cells

To ensure optimal growth of strain GtCom6, it was sub-cultured for three generations on pre-warmed TSA plate at 60 °C. 3–4 single colonies were selected as inoculum for 6 mL of pre-warmed liquid mLB medium in small sterile plugged glass test tube with dimensions of 1.8 × 20 cm. The Cells were then cultivated at 60 °C with shaking at 250 rpm for ~10 h to produce a seed liquid. Afterward, ~5 ml of the seed solution was transferred into a 250 mL conical flask that already had 50 mL of mLBS medium (adding 0.5 M sorbitol to mLB), and the solution was then adjusted to an $OD_{600}$ of 0.1. The flask was subsequently placed in the incubator set to 60 °C and 250 rpm, and left until the $OD_{600}$ level reached 1.8 (± 0.1). Glycine (2%), DL-alanine (1%), and Tween 80 (0.0375%) were added to the flask, and the cells were cultured for an additional hour. The flask was then cooled on ice for 10 min before the cells were harvested by centrifugation at 4 °C for 20 min at 8000 × *g*. The cells were washed 4–5 times with 15 ml of ice-cold SMGT medium (0.5 M Sorbitol; 0.5 M Mannitol; 10% v/v Glycerol; 0.5 M Trehalose) and the final pellets were resuspended in 800 μL of ice-cold SMGT solution. Finally, 80 μL aliquots were transferred to 1.5 mL EP tubes and stored at −80 °C.

The process of generating GtCom7 competent cells was closely resembled that of Gtcom6, except that 15 mM xylose was added to induce CRISPR array expression during the transfer of the seed solution to the conical flask.

## Testing high-efficiency electrotransformation-competent cell

To assess the transformation efficiency, 80 μL of competent cells were combined with 1 μL (0.5 ng/μL) of plasmid pUCG3.8 and loaded into a precooled 1 mm gap electroporation cuvette placed on ice. Electroporation was performed using a Bio-Rad gene pulser apparatus (Bio-Rad Laboratories, Richmond, CA) under the following conditions: 2.5 kV, 25 μF, and 200 Ω, with a time constant greater than 5 ms. Immediately following electroporation, the cells were transferred to a 50 ml falcon tube containing 1 ml of pre-warmed TGP at 52 °C and incubated with agitation at 200 rpm for 2–4 h. The cell suspension was then serially diluted in pre-warmed TGP and plated directly onto TSA supplemented with Km (12.5 μg/ml). The plates were incubated at 52 °C for 24 h, and the transformation efficiency of competent cells of strain was determined by calculating the number of transformants obtained per microgram of plasmid DNA.

## Design and synthesis of genome-scale tgRNA library

The *P. thermoglucosidasius* NCIMB 11955 genome sequence and open reading frame (ORF) annotations of 3844 genes were obtained from National Center for Biotechnology Information (GenBank, CP016622.1), which was used for the tgRNA library (20-mer) design. Briefly, three tgRNAs were chosen to target the template strand of each gene, positioned at the upstream, middle, and downstream regions. For tgRNAs generated, python codes were used for design (Supplementary Note 2). The final library contains 11281 tgRNA sequences targeting 3821 genes (Supplementary Data 2). Priming sequences and BsaI restriction sites were added to the 5′ and 3′ ends of each tgRNA for PCR amplification and Golden Gate assembly. The final designed oligonucleotide library was synthesized on a chip by GenScript company (Jiangsu, China).

## Construction of tgRNA plasmid library

The repression plasmid pZH04 with BsaI sites in this study was used for construction of tgRNA library. The oligonucleotide library was cloned into pZH04 with BsaI sites using Golden Gate assembly by GenScript company. The plasmid library obtained from GenScript company was sequenced by NGS to assess the plasmid library quality.

## Development of the CRISPRi library strain and competent cells

To obtain a CRISPRi library strain of GtCom6, 2 μL (approximate 500 ng/μL) library plasmid was electrotransformed into high-efficiency competent cells of strain GtCom6. The resulting cells were then spread onto a TSA plate that had been supplemented with Km. The plate was incubated at 52 °C until colonies appeared, after which the GtCom6 strains carrying the CRISPRi plasmid library were obtained by washing all the colonies from the plate with ~10 ml of LB liquid medium. Then the GtCom6 library strains were prepared as competent cells by the method above.

## Screening positive targets involved in transformation

To reduce the negative colonies of Gtcom6 library strains following electroporation with test plasmid pZH04-spe, a two-step approach was taken. Firstly, the optimal amount of pZH04-spe to be added to the competent cells of GtCom6 library strains was determined. Different amounts of plasmids were electrotransformed into the competent cells of Gtcom6 strain (not Gtcom6 library strain) and cultured on plates supplemented with Sp. If 1−9 colonies were able to grow on the plate, the amount of test plasmid added to the electrotransformation-competent cell was deemed optimal. Secondly, the desired amount of pZH04-spe was mixed with competent cells of Gtcom6 library strains and electrotransformed (repeated 10 times). The resulting strains were cultured on the ten TSA agar plates supplemented with Sp, and all the strains that grew on these plates were collected and mixed together as a template for PCR. NGS and processing was carried out by GenScript company (Jiangsu, China). Abundance changes of tgRNAs assayed by deep sequencing, and the top 6 significantly enriched tgRNAs were chosen for further experimental validation (Supplementary Data 5).

## Statistics and reproducibility

The data in all Figures are expressed as means ± standard deviation (SD) with statistical significance assessed by Student's *t* test. Microsoft Excel 2016, and GraphPad Prism version 8.4.0 were used for the data analysis. Each figure includes data from one representative biological replicate of the experiment. Unless otherwise specified in the figure legends, all experiments were independently conducted three or more times under either identical or closely similar conditions.

## Reporting summary

Further information on research design is available in the Nature Portfolio Reporting Summary linked to this article.

## Data availability

The authors declare that the main data supporting the findings of this study are available within the paper and its Supplementary Information files. The RNA-seq data of *P. thermoglucosidasius* strain used in this study are available in the NCBI Short Read Archive with BioProject ID PRJNA1018439. Source data are provided with this paper, and extra data are available from the corresponding author upon request.

## Code availability

The integrated software used for tgRNA library design can be found at https://zenodo.org/record/8351358 (https://doi.org/10.5281/zenodo.8351358).

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

## Acknowledgements

This work was supported by the National Key Research and Development Program of China (2020YFA0906800 to H.X. and Z.L.); the National Natural Science Foundation of China (grants 32150020 to M.L., 32170095 to W.W., and 32100066 to H.Y.); the Youth Innovation Promotion Association CAS (Y202027 to W.W.); and 2023 Double World-class Project-Key Program-Intelligent Biomanufacturing to L.Z.

## Author contributions

W.W., Z.Y., Z.L., and L.Z. conceived and supervised the project. Z.Y., B.L., and R.B. designed and performed the main experiments. Z.W. and Z.X. performed the bioinformatics analyses. H.Y. participated in the experiments. Z.Y. and W.W. wrote the manuscript. M.L., H.X., G.T., and G.Z. gave feedback on the manuscript.

## Competing interests

The authors have filed a provisional patent for this work to the China National Intellectual Property Administration (CNIPA: 202310705228X). W.W., Z.Y., Z.L., and R.B. are inventors on the provisional patent application. The remaining authors declare no competing interests.
