## [Peer Review File · Nature Communications]

Reviewers' Comments:

Reviewer #1:

Remarks to the Author:

In this manuscript, authors identified a thermophilic type I-B CRISPR-Cas system from *P. thermoglucosidasius*, and the nuclease-based cleavage can be switched to transcriptional repression using this type I-B system by adjusting crRNA length to less than 26 bp without ablating Cas3 nuclease. This endogenous system is very convenient for genetic modification of the strains. And using this type I-B system, dynamically controlled supercompetent *P. thermoglucosidasius* cells with unprecedented efficiency ($\sim 10^8$ CFU/ μ g DNA) by temporal multiplexed repression were established. The authors addressed two major obstacles, high-efficient genetic manipulation and transformation, that hindered the broader understanding and engineering of this species. In this regard, the current work can provide an interesting tools and strategy to readers. However, there are several concerns need to be clarified and addressed before publication.

- 1) The introduction to the principle of CRISPR-Cas is too little and needs to be supplemented.
- 2) line 262, have you tested any other genes besides sfGFP? What are the excitation and emission wavelengths of fluorescence determination? Please add them to the materials and methods.
- 3) Full text abbreviations should be marked in full when they first appear.
- 4) In 450 lines, "To identify more unknown targets, a genome-scale CRISPRi library was used to rapidly screening based on the observable colorimetric phenotype of riboflavin", What is the storage capacity of CRISPRi library? Is this CRISPRi library the same as the library when improving conversion efficiency (Lines 373, 374, and 375) ?
- 5) In line 457: Sometimes the empty vector of CRISPRi plasmid has an effect on riboflavin production, so we think the control strain should be Gt-08 carried empty vector rather than the Gt-08 in flask fermentation.
- 6) On the plate, is the riboflavin produced by strain *P. thermoglucosidasius* an intracellular or extracellular product? Identifying the high and low Production of Riboflavin by Colony Color, Will the selected strains only be those with high intracellular riboflavin production?
- 7) There are some unclear contents on the fermentation in bioreactor. The parameters and control methods in the fermentation process in the bioreactor are missing. For example, the initial agitation speed, air flow, ratio of inoculation. Is the fermentation was perform by fed-batch? If yes, please add the information of the feed medium. Sometime the improvement of titer in flask are not reproducible in bioreactor, so the production of the control strain in the bioreactor are needed. When is the sugar exhausted in the flask fermentation? And how long was the fermentation process?
- 8) Figure 6F shows that although OD600 began to decrease after 16 hours of fermentation, the production of riboflavin has been increased. Why did the fermentation stop after 28 hours and not continue to extend the fermentation time? What is the yield (conversion rate) of riboflavin/glucose? Which should be described and discussed in the manuscript.
- 9) Change 27.6% in the "Tier% increase" column of "Supplementary Table 8" to 27.6.

Reviewer #2:

Remarks to the Author:

In their manuscript, 'A thermostable type I-B CRISPR-Cas system for orthogonal and multiplexed genetic engineering', Yang et al. identify a type I-B CRISPR-Cas system in the thermophilic bacterium *Parageobacillus thermoglucosidasius*. They demonstrate that this system, like it was previously shown for other type I CRISPR-Cas systems, loses its DNA cleaving activity upon reduction of the crRNA length but its DNA binding capacity is retained. Based on their findings, they employ the I-B system to develop orthogonal genome editing and transcriptional repression tools in *P. thermoglucosidasius* and *E. coli*. Further on, they increase the transformation efficiency of *P. thermoglucosidasius* employing a genome-scale screening approach to identify genes involved in the transformation efficiency of the strain, beyond its R-M systems, and downregulate them in a dynamic and combinatorial manner. Finally, they employ the same genome scale approach to turn *P. thermoglucosidasius* into a thermophilic riboflavin cell factory.

Overall, the manuscript is well-written and there are many interesting data-sets. Nonetheless, in my opinion, the authors make some claims that are not sufficiently supported by the provided data. Moreover, the manuscript lacks novelty to warrant publication in Nature Communications. Please see below my major and minor points of concern:

Major points

Lines 217-233: The authors emphasize the specificity of the here reported I-B system. One of the main arguments is its very strict PAM requirement. Nonetheless, the described PAM identification process is limited to a very small number of tested motifs. Even if it is hypothesized that the PAM of the system is 3 nucleotides long, like it is for most other type I CRISPR-Cas systems, there are 64 possible tri-nucleotide combinations for such a motif. The authors only test 10 of them, due to their assumption that 2 out of 3 nucleotides from the initial TTA PAM should always remain the same (Figure 3c). This is not a correct assumption, since there are many examples of systems that recognize more diverse PAMs (f.e. the *E. coli* I-E system recognizes ATG, AAG, AGG, GAG and TAG as PAMs - PMID: 26863189). Hence, the authors should follow a more detailed PAM identification process, as f.e. described in (PMID: 32246713).

Line 258: In continuation to the above comment, the fact that prime adaptation shows preference for the TTA PAM does not mean that there are no other less preferred PAMs. The identification of these less preferred PAMs is important for claims regarding the specificity of the system.

Lines 369-396: The here described genome-scale screening approach, which is based on >11.000 gRNAs, can be used with strains that already have high-enough transformation efficiency, like the 10^6 CFU/ug efficiency of the strain that the authors used. Hence, this approach cannot be used for the vast majority of the non-model bacteria with extremely low transformation efficiencies. Moreover, the authors should make sure (via total RNA-seq) that the 6 identified tgRNAs do not downregulate the expression of other genes (off-targets) as it has been reported for other systems (f.e. PMID: 29946130).

Line 456: The authors should reveal the identity of the identified targets and discuss how their downregulation increases riboflavin production. Moreover, as suggested in my comment above, the authors should make sure that the identified tgRNAs do not downregulate the expression of other genes.

Minor points:

Lines 61-63: The authors here claim that since *Haloarcula hispanica* is a halophile then its I-B CRISPR-Cas system should not be active in a non-halophilic organism. Do the authors expect that this I-B system requires high salinity conditions to function? Are there any data that support this claim?

Line 490-493: The authors should provide references that specifically support their claims regarding the toxicity and the off-target activity of the mentioned Cas9 orthologs. The provided references do not clearly support their claims.

Supplementary Figure 5c: How many times were these experiments performed? The corresponding graphs do not contain standard deviations, which are important for the statistical significance of these results.

Reviewer #3:

Remarks to the Author:

* Reviewer Comments to Author:

The authors identified a type I-B CRISPR-Cas system from thermophilic bacterium *Parageobacillus thermoglucosidasius*, and developed it as both the highly efficient genome editing and CRISPRi system, only by switching crRNA length (>26 bp for genome editing, ≤26bp for transcriptional repression). This Cas system was also effective in mesophilic bacterium *E. coli*. By employing this system for genome-scale CRISPRi screening, the target genes for improving transformation efficiency and riboflavin titer was found. By dynamic controlling these genes through xylose-

inducible expression of crRNAs, the authors achieved a dramatic increase in transformation efficiency (approximately 10^8 CFU/ug DNA) and obtained the hitherto highest titers of riboflavin in high-temperature fermentation. This study provides clear results and offers efficient genetic tools for thermophilic bacteria, which have recently spotlighted for their industrial potential. However, several additional experiments or discussions may be required.

Comments

1. As the authors described in the introduction section, Cas effectors from halophilic organism are prohibitive to the other large majority of microorganisms due to the high salinity requirements for activity and stability. In contrast, thermostable type I-B CRISPR-Cas of this study is also well-performed in mesophilic *E. coli*. Please discuss about the potential biological or structural reasons of less requirements for activity and stability of this Cas system, and the perspectives of the universal application of this Cas system to other bacteria.
2. Is the Cas3-based genome editing efficiency or time-scale better than previous double-crossover based method in *P. thermoglucosidasius*?
3. In line 111, Supplementary table 3 is appeared first before Supplementary table 1 and 2.
4. In Supplementary Figure 1, please indicate the phylogenetic tree information at the figure legend.
5. In line 234-235, the sentence should be revised.
6. In Supplementary Figure 4, the description about 4B and 4C was absent in the Result section.
7. In Figure 5D, it may be better to represent enriched tgRNAs as the relative read count fold change compared to initial competent cell library in Figure 5C, rather than the number of read count, to avoid the bias generated from the initial plasmid library or the competent cell library.
8. In line 402-417, the repression efficiency of each five tgRNA in the multiplexing array according to their order may be tested.
9. In line 450-466, please clarify how genome-scale CRISPRi library was introduced to Gt-08 strain which seems to already have multiplex repression CRISPRi vector (five genes for enhancing competency).
10. In line 450-466, are the identified target genes by CRISPRi for enhancing riboflavin titers could be predicted by the genome-scale metabolic model (<https://doi.org/10.1016/j.ymben.2021.03.002>)?
11. A previous report (<https://doi.org/10.1021/acssynbio.1c00138>) in the same species developed thermostable CRISPR-Cas9 system. Please emphasize or provide the evidence for the novelty or efficiency of the type I-B CRISPR-Cas system of this study compared to the Cas9 system, in addition to the advantages of PAM described in discussion section (line 493-497).
12. In line 557, "thespacer" should be "the spacer".
13. In Supplementary Note 2, please describe the design principle of tgRNA library, including rationale for avoiding off-targets. How was the top three tgRNAs selected for each gene?

What are the noteworthy results?

- The authors identified a type I-B CRISPR-Cas system from thermophilic bacterium *Parageobacillus thermoglucosidasius*, and developed it as both the highly efficient genome editing and CRISPRi system, only by switching crRNA length (>26 bp for genome editing, ≤26bp for transcriptional repression). This Cas system was also effective in mesophilic bacterium *E. coli*. By employing this system for genome-scale CRISPRi screening, the target genes for improving transformation efficiency and riboflavin titer was found. By dynamic controlling these genes through xylose-inducible expression of crRNAs, the authors achieved a dramatic increase in transformation efficiency (approximately 10^8 CFU/ug DNA) and obtained the hitherto highest titers of riboflavin in high-temperature fermentation.

Will the work be of significance to the field and related fields?

- This study provides clear results and offers efficient genetic tools for thermophilic bacteria, which have recently spotlighted for their industrial potential. However, it is quite limited to the specific bacteria strain, that may be not interested by broad-audience.

How does it compare to the established literature? If the work is not original, please provide relevant references

- It is original.

Does the work support the conclusions and claims, or is additional evidence needed?

- Several additional experiments or discussions may be required. (Please see comments above)

Are there any flaws in the data analysis, interpretation and conclusions? Do these prohibit publication or require revision?

- No.

Is the methodology sound? Does the work meet the expected standards in your field?

- Most of data and methodology are sufficient, except for some aspects (please see comments)

Is there enough detail provided in the methods for the work to be reproduced?

- Yes

Point-by-point response

Dear editor and reviewers,

On behalf of my co-authors, we greatly appreciate the constructive comments and suggestions from the editor and the reviewers.

The comments of the reviewers are all valuable and helpful for improving our manuscript. We have fully addressed all the comments, and listed the point-by-point response as following.

Reviewers' comments:

Reviewer #1 (Remarks to the Author):

In this manuscript, authors identified a thermophilic type I-B CRISPR-Cas system from *P. thermoglucosidasius*, and the nuclease-based cleavage can be switched to transcriptional repression using this type I-B system by adjusting crRNA length to less than 26 bp without ablating Cas3 nuclease. This endogenous system is very convenient for genetic modification of the strains. And using this type I-B system, dynamically controlled supercompetent *P. thermoglucosidasius* cells with unprecedented efficiency ($\sim 10^8$ CFU/ μ g DNA) by temporal multiplexed repression were established. The authors addressed two major obstacles, high-efficient genetic manipulation and transformation, that hindered the broader understanding and engineering of this species. In this regard, the current work can provide an interesting tools and strategy to readers. However, there are several concerns need to be clarified and addressed before publication.

1) The introduction to the principle of CRISPR-Cas is too little and needs to be supplemented.

Our response:

Many thanks for your suggestion.

We have added more information about principle of CRISPR-Cas in the instruction.

2) line 262, have you tested any other genes besides sfGFP? What are the excitation and emission wavelengths of fluorescence determination? Please add them to the materials and methods.

Our response:

Thank you for your comments.

Line 262: In addition to sfGFP, we have also tested another two genes, namely *purA* (BCV53_03205) and *purR* (BCV53_02825). The three demonstrations together provided comprehensive evidences that transcription repression did not impair when deleting Cas1-2 and Cas4. These results were shown in the additional Figure 1.

As for excitation and emission wavelengths used in this work, the wavelength of 488 nm and 510 nm were used, respectively. We have included this information in the Materials and Methods section to ensure clarity and repeatability.

Additional Figure 1. (A) Testing the transformation efficiency of plasmid pPurA-T30 and pPurR-T30, both contain a 30-bp truncating spacer, in strain Y2 and Y3. Strain Y2 and Y3 containing control plasmid pZH04 as control strain. (B) Evaluating the transcription repression ability of plasmid pPurA-T25 and pPurR-T25, both contain a 25-bp truncating spacer, on PurA and PurR protein in the genomes of strain Y2 and Y3. Strain Y2 and Y3 containing control plasmid pZH04 as control strain. Data are the mean of three biological repeats and are expressed as mean \pm SD.

3) Full text abbreviations should be marked in full when they first appear.

Our response:

Thank you for your kind remind.

We have re-checked the paper and spelled out all abbreviations completely when they are first mentioned.

4) In 450 lines, “To identify more unknown targets, a genome-scale CRISPRi library was used to rapidly screening based on the observable colorimetric phenotype of riboflavin”, What is the storage capacity of CRISPRi library? Is this CRISPRi library the same as the library when improving conversion efficiency (Lines 373, 374, and 375) ?

Our response:

Thank you for your comments.

In this work, the CRISPRi library mentioned in lines 450 and lines 373-375 is the same one, which harbored 11281 crRNA sequences. This library is qualified for rapid screening of unknown targets of diverse phenotypes of *P. thermoglucosidasius*, including here showed the improvement of riboflavin production and transformation efficiency.

5) In line 457: Sometimes the empty vector of CRISPRi plasmid has an effect on riboflavin production, so we think the control strain should be Gt-08 carried empty vector rather than the Gt-08 in flask fermentation.

Our response:

Thank you for your comment.

Actually, the strain Gt-08 used in flask fermentation carries the pZH04 empty vector. We apologize for not describing this information in the original manuscript. We sincerely appreciate your carefulness that help us to improve the accuracy of our manuscript.

6) On the plate, is the riboflavin produced by strain *P. thermoglucosidasius* an intracellular or extracellular product? Identifying the high and low Production of Riboflavin by Colony Color, Will the selected strains only be those with high intracellular riboflavin production?

Our response:

Thank you for your comments.

Riboflavin produced by *P. thermoglucosidasius* is primarily as an extracellular product. Also, previous reports revealed that riboflavin was produced and secreted outside the cell by both *Bacillus subtilis*¹ (PMID: 37110496) and *Yarrowia lipolytica*² (PMID: 29698778).

We agree with you that that colony color may not always directly correlate with extracellular riboflavin production. In our study, we initially select strains with high riboflavin production by visible color of clones on plates. After visual screening, we further confirm their riboflavin production levels by evaluating the riboflavin content in the submerge fermentation.

We compared the riboflavin titer in the fermentation supernatant of the selected strains with a control strain Gt-08, which contains an empty vector plasmid. Our results showed that the riboflavin titer in the fermentation supernatant of the selected strains was significantly higher than that of the control strain Gt-08 (Figure 6D and Supplementary Figure 12A).

7) There are some unclear contents on the fermentation in bioreactor.

The parameters and control methods in the fermentation process in the bioreactor are missing. For example, the initial agitation speed, air flow, ratio of inoculation. Is the fermentation was perform by fed-batch? If yes, please add the information of the feed medium.

Sometime the improvement of titer in flask are not reproducible in bioreactor, so the production of the control strain in the bioreactor are needed.

When is the sugar exhausted in the flask fermentation? And how long was the fermentation process?

Our response:

Thank you for your comments.

Parameters and Control Methods have been added as follow:

Seed cultures for the fermenters were prepared by inoculating 5 colonies from a plate into 100 mL of seed medium in a 500 ml flask. The flask was cultured at 60 °C and agitated at 250 rpm for 14 h. Fermentation were carried out in 5-L fermentation bioreactor (Bailun Bio, Shanghai, China) with 2.5 L medium. And the inoculation volume was 5% (v/v). The bioreactor was agitated at 600 rpm, and air supplied at a rate of 2 L/min. The fermentation was run at 60 °C and pH of 7.0.

We indeed performed a fed-batch fermentation process, where glucose was added to the bioreactor to maintain a suitable sugar concentration between 0 and 5 g/L.

As for the glucose consumption, 20 g/L was exhausted within 10 h in the flask fermentation. The fermentation process lasted 48 h. In parallel, the same initial glucose concentration of 20 g/L was exhausted within 12 h in the bioreactor by process control, and the duration time was 28 h, as demonstrated in Figure 6F.

8) Figure 6F shows that although OD₆₀₀ began to decrease after 16 hours of fermentation, the production of riboflavin has been increased. Why did the fermentation stop after 28 hours and not continue to extend the fermentation time? What is the yield (conversion rate) of riboflavin/glucose? Which should be described and discussed in the manuscript.

Our response:

Thank you for your comments.

Due to the highest titer appeared at 28 hours in our optimization process, we did not to show the result with more extended time.

The yield is 2.1 g riboflavin/100 g glucose, which was lower than what has been reported for *Bacillus subtilis* and *E. coli* (PMID: 37110496, PMID: 34623820). One possible reason for the lower yield in strain Gt-09 is the production of 10 g/L acetic acid in flask fermentation. Additionally, we did not specially optimize for the yield in strain Gt-09. As a result, future research should focus on further manipulating the strain through rational engineering to achieve a higher yield.

Given the aim of constructing thermophilic riboflavin cell factory was to showcase the high efficiency of our genetic tool, which enabled rapid and multiplexed genetic engineering, including knockout, knock-in, point mutation and multiplexed gene repression (as shown in Figure 6A), we did not discuss and compare more about the titer, yield, and productivity.

9) Change 27.6% in the "Titer% increase" column of "Supplementary Table 8" to 27.6.

Our response:

Thank you for your careful review.

We have revised as you suggested in the Supplementary Table 8.

Reviewer #2 (Remarks to the Author):

In their manuscript, ‘A thermostable type I-B CRISPR-Cas system for orthogonal and multiplexed genetic engineering’, Yang et al. identify a type I-B CRISPR-Cas system in the thermophilic bacterium *Parageobacillus thermoglucosidasius*. They demonstrate that this system, like it was previously shown for other type I CRISPR-Cas systems, loses its DNA cleaving activity upon reduction of the crRNA length but its DNA binding capacity is retained. Based on their findings, they employ the I-B system to develop orthogonal genome editing and transcriptional repression tools in *P. thermoglucosidasius* and *E. coli*. Further on, they increase the transformation efficiency of *P. thermoglucosidasius* employing a genome-scale screening approach to identify genes involved in the transformation efficiency of the strain, beyond its R-M systems, and downregulate them in a dynamic and combinatorial manner. Finally, they employ the same genome scale approach to turn *P. thermoglucosidasius* into a thermophilic riboflavin cell factory.

Overall, the manuscript is well-written and there are many interesting data-sets. Nonetheless, in my opinion, the authors make some claims that are not sufficiently supported by the provided data. Moreover, the manuscript lacks novelty to warrant publication in Nature Communications. Please see below my major and minor points of concern:

Response:

Thank you for your comments.

As for the novelty and the degree of advance, to our knowledge, the technique that using intact type I CRISPR-Cas for orthogonal genome editing and transcription repression by adjusting crRNA length has not developed in both thermophile and mesophile hosts.

After the discovery that Type I-B Cas effectors can be guided by non-canonical crRNAs to mediate transcriptional repression without ablating Cas3 nuclease in *Haloarcula hispanica*³, Li et al demonstrated the roles of simultaneous gene regulation and editing in this host⁴. However, these proteins were conducted in a halophilic organism with high salinity requirements for activity and stability that are prohibitive to the large majority of microorganisms. We here sought to characterize and harness an ideal thermostable type I CRISPR-Cas system to address current limitations in the efficiency of genetic manipulation tools for engineering thermophilic and mesophilic microbial factories. Given the widely engineering of mesophilic and thermophilic microbial factories, we believe this work have potential

interest to researchers.

Especially, high temperature fermentation offers a number of advantages over mesophilic biorefineries, including faster feedstock conversion rates, reduced bioreactor cooling costs, lower risk of contamination, and decreased viscosity during fermentation. Increasing temperature can also make unfavorable reaction in mesophiles thermodynamically feasible. Our work enables high-efficiency genetic manipulation in *P. thermoglucosidasius* and facilitates the engineering of thermophilic cell factories. We therefore believe our work will contribute to an upsurge in the engineering of *P. thermoglucosidasius* as an alternative and attractive host for thermophilic cell factories.

To demonstrate the usefulness and potentiality, we demonstrated three breakthrough practical applications in this work, namely, a novel strategy for screening the global targets involved in transformation efficiency, a dynamically controlled super-competent with unprecedented efficiency ($\sim 10^8$ CFU/ μ g DNA), and a construction of thermophilic riboflavin cell factory with hitherto highest titers at high-temperature fermentation condition.

Some highlights of this exciting work are again summarized below:

1. We characterized a thermophilic type I-B CRISPR-Cas system, then harnessed it for orthogonal gene knockout and transcriptional repression by adjusting the length of crRNAs, without ablating Cas3 nuclease.
2. This orthogonal tool could concurrently perform DNA edits and transcriptional repression with high efficiency and accuracy in both thermophile and mesophile hosts. To our knowledge, it is the first time to develop such orthogonal and multiplexed type I-B genetic tool in both thermophile and mesophile hosts.
3. Using this tool, we designed a novel workflow for enhancement of transformation efficiency, including the iterative deletion the known targets, the genome-scale screening the known targets, and the establishment of a dynamically controlled supercompetent. We finally achieved the transformation of *P. thermoglucosidasius* cells with unprecedented efficiency ($\sim 10^8$ CFU/ μ g DNA). To our knowledge, it is the first time to develop dynamically controlled transformation-competent cell.
4. Empowered by the advances of efficient genetic manipulation and transformation efficiency, we were able to construct a riboflavin overproduction cell factory with hitherto highest titers in high temperature fermentation. The overall time we spent is about two months.

Major points

Lines 217-233: The authors emphasize the specificity of the here reported I-B system.

One of the main arguments is its very strict PAM requirement. Nonetheless, the described PAM identification process is limited to a very small number of tested motifs. Even if it is hypothesized that the PAM of the system is 3 nucleotides long, like it is for most other type I CRISPR-Cas systems, there are 64 possible tri-nucleotide combinations for such a motif. The authors only test 10 of them, due to their assumption that 2 out of 3 nucleotides from the initial TTA PAM should always remain the same (Figure 3c). This is not a correct assumption, since there are many examples of systems that recognize more diverse PAMs (f.e. the E. coli I-E system recognizes ATG, AAG, AGG, GAG and TAG as PAMs - PMID: 26863189). Hence, the authors should follow a more detailed PAM identification process, as f.e. described in (PMID: 32246713).

Our response:

Thank you for your comments.

As you suggested, we have carried out a more detailed PAM identification process.

We have now expanded our investigation to include all 64 possible tri-nucleotide combinations for the PAM motif. The results are shown in the following Figure, as well as in the revised Figure 3C. By determining the interference efficiency, we found that TTA is the best PAM. We also observed that the PAM of CTA, TCA, TTG, and TTT displayed interference activity, but the efficiency is two orders of magnitude lower compared with that of TTA.

In our study, we focused on determining the specific transcription repression capabilities of the type I-B CRISPR-Cas system with the best PAM of TTA. Therefore, we did not pay more attention on those imperfect PAMs.

Revised Figure 3. (C) Effects of 64 possible tri-nucleotide PAM motifs on interference efficiency.

Line 258: In continuation to the above comment, the fact that prime adaptation shows preference for the TTA PAM does not mean that there are no other less preferred PAMs. The identification of these less preferred PAMs is important for claims regarding the specificity of the system.

Our response:

Thank you for your comments.

To address your concern of the less preferred PAMs, we have conducted additional experiments to identify potential less preferred PAM sequences (Revised Figure 3C).

We found the interference efficiency of these less preferred PAMs (CTA, TCA, TTG, and TTT) is two orders of magnitude lower than that of TTA. These less preferred PAMs are not qualified for tool development. Therefore, we did not evaluate the prime adaptation of these less preferred PAMs.

In our study, our main goal was to develop an orthogonal and multiplexed genetic engineering tools using this thermostable type I-B CRISPR-Cas. We therefore characterized the performance of prime adaptation for the qualified TTA PAM motif, which showed the best specific PAM.

Lines 369-396: The here described genome-scale screening approach, which is based on >11.000 gRNAs, can be used with strains that already have high-enough transformation efficiency, like the 10^6 CFU/ug efficiency of the strain that the authors used. Hence, this approach cannot be used for the vast majority of the non-model bacteria with extremely low transformation efficiencies.

Our response:

Thank you for your comments.

We agree with you that the genome-scale screening approach cannot be used for the vast majority of the non-model bacteria with extremely low transformation efficiencies. Therefore, we highlighted the workflow we developed for improving the transformation efficiency (Figure 5A). In the first step, we introduced iterative deletions to remove the putative endogenous defense systems. The aim of this initial step is to elevate the transformation efficiency to 10^6 CFU/ug. To further improve the efficiency, genome-scale screening approach was carried out. We think the workflow may be helpful to some of the non-model bacteria.

As our demonstrated, the original transformation efficiency of *P. thermoglucosidasius* was only 2×10^3 CFU/ μ g DNA, which is unmet the requirement of genome-scale screening. It is right the case you concerned. Through our three-step workflow, we constructed super-competent with unprecedented efficiency ($\sim 10^8$ CFU/ μ g DNA).

Moreover, the authors should make sure (via total RNA-seq) that the 6 identified tgRNAs do not downregulate the expression of other genes (off-targets) as it has been reported for other systems (f.e. PMID: 29946130).

Our response:

Thank you for your comments.

Usually, reporter genes were used for evaluation of off-targets, because they do not have physiological function. For example, Mikko Taipale et al. assess the potential off-target of CRISPRi using EGFP in human cells⁵ (PMID: 33020655). Here, we also demonstrated the specificity of transcription repression by targeting the reporter gene *gfp*. Therefore, the specificity of transcription repression had been confirmed.

We are unable to measure the off-targets the 6 identified tgRNAs. Because we do not judge the downregulated genes are caused by the repression of the 6 targets or the off-

target of 6 tgRNAs. This means it is not known whether the downregulated genes are the direct cause of off-target. If we assessed off-target of the 6 identified tgRNAs, we must make sure that the 6 identified tgRNAs would have few downstream targets that could confound specificity analysis⁶ (PMID: 25494202). Actually, the 6 identified tgRNAs greatly influenced the physiological state, including the reduced growth (Supplementary Figure 10F) and the elevated transformation efficiency (Figure 5B).

We also scan the reference⁷ (PMID: 29946130) you mentioned, it presents a method that enables investigation of the cellular consequences of repressing individual transcripts based on the CRISPR interference (CRISPRi) pooled screening, and do not test that whether the identified gRNAs downregulate the expression of other genes.

Line 456: The authors should reveal the identity of the identified targets and discuss how their downregulation increases riboflavin production. Moreover, as suggested in my comment above, the authors should make sure that the identified tgRNAs do not downregulate the expression of other genes.

Our response:

Thank you for your comments.

As you suggested, we have listed the identified targets and their functions in the revised supplementary Table 6.

As for the off-target of the developed tool, we have fully explained in the above response. Here, we used a schematic to explain why we did not design experiment as you suggested. Because we do not judge the downregulated genes are caused by the repression of the identified targets (A route) or the off-target of tgRNAs (B route).

Additional Figure 2. Schematic of the relationships between tgRNA and the affected genes.

Minor points:

Lines 61-63: The authors here claim that since *Haloarcula hispanica* is a halophile then its I-B CRISPR-Cas system should not be active in a non-halophilic organism. Do the authors expect that this I-B system requires high salinity conditions to function? Are there any data that support this claim?

Our response:

Thank you for your comments.

Proteins from halophile require high salinity for activity and stability that are prohibitive to the large majority of microorganisms⁸ (PMID: 31877629). Likewise, our developed orthogonal and multiplexed tool is unable to be used in high salinity conditions. It can be used in both thermophile and mesophile hosts for genome editing and transcriptional repression. Given the widely engineering of mesophilic and thermophilic microbial factories, we believe our developed tool have potential interest to researchers for orthogonal and multiplexed genetic engineering.

Line 490-493: The authors should provide references that specifically support their claims regarding the toxicity and the off-target activity of the mentioned Cas9 orthologs. The provided references do not clearly support their claims.

Our response:

Thank you for your comments.

The reference⁹ (PMID: 31890588) have been provided in the revised version.

Supplementary Figure 5c: How many times were these experiments performed? The corresponding graphs do not contain standard deviations, which are important for the statistical significance of these results.

Our response:

Thank you for your comment.

As you suggested, we have revised Supplementary Figure 5C to include standard deviations. The data in the figure represent three biological replicates.

Reviewer #3 (Remarks to the Author):

* Reviewer Comments to Author:

The authors identified a type I-B CRISPR-Cas system from thermophilic bacterium *Parageobacillus thermoglucosidasius*, and developed it as both the highly efficient genome editing and CRISPRi system, only by switching crRNA length (>26 bp for genome editing, ≤26bp for transcriptional repression). This Cas system was also effective in mesophilic bacterium *E. coli*. By employing this system for genome-scale CRISPRi screening, the target genes for improving transformation efficiency and riboflavin titer was found. By dynamic controlling these genes through xylose-inducible expression of crRNAs, the authors achieved a dramatic increase in transformation efficiency (approximately 10^8 CFU/ug DNA) and obtained the hitherto highest titers of riboflavin in high-temperature fermentation. This study provides clear results and offers efficient genetic tools for thermophilic bacteria,

which have recently spotlighted for their industrial potential. However, several additional experiments or discussions may be required.

Comments

1. As the authors described in the introduction section, Cas effectors from halophilic organism are prohibitive to the other large majority of microorganisms due to the high salinity requirements for activity and stability. In contrast, thermostable type I-B CRISPR-Cas of this study is also well-performed in mesophilic *E. coli*. Please discuss about the potential biological or structural reasons of less requirements for activity and stability of this Cas system, and the perspectives of the universal application of this Cas system to other bacteria.

Our response:

Thanks for your comments.

Proteins from halophilic organisms are characterized by their high content of acidic amino acids (Aspartate and glutamic acid), being only stable in solutions containing high salt concentration (between 1 and 4 M total salt concentration)⁸ (PMID: 31877629).

Thermophilic proteins, such as type I-B CRISPR-Cas of *P. thermoglucosidasius*, maintain a thermodynamically stable fold in both thermophile and mesophile hosts. The extended activity and stability of this CRISPR-Cas allows for its application in molecular biology techniques that require DNA manipulation at temperatures of 20–70 °C, as well as its exploitation in harsh environments that require robust protein activity.

2. Is the Cas3-based genome editing efficiency or time-scale better than previous double-crossover based method in *P. thermoglucosidasius*?

Our response:

Thank you for your valuable comments.

Yes, the Cas3-based genome editing method exhibits superior efficiency and time-scale compared to the previous double-crossover based method in *P. thermoglucosidasius*. The maximum editing efficiency of the double-crossover based method in *P. thermoglucosidasius*, theoretically, is 50%, and some studies have reported gene editing efficiencies close to 50%^{10, 11} (PMID: 28066509 and 32096925). Additionally, the double-crossover based method often requires the use of a counter-selection marker to improve the efficiency of screening positive colonies. Thus additional time are required to obtain the counter-selection marker.

On the other hand, the Cas3-based gene editing method has demonstrated nearly 100% editing efficiency, as shown in Figure 4D of our study. Furthermore, it does not require the use of a selection marker, simplifying the editing process and reducing the time needed for successful editing.

3. In line 111, Supplementary table 3 is appeared first before Supplementary table 1 and 2.

Our response:

Thank you for carefully review.

We have revised the order of the tables in the supplementary file.

4. In Supplementary Figure 1, please indicate the phylogenetic tree information at the figure legend.

Our response:

Thank you for your comments.

We have updated the figure legend for Supplementary Figure 1 to include the following information:

Phylogenetic tree constructed by neighbor-joining algorithm from the distance matrix of 16S rDNA partial sequences of thermophilic *Geobacillus* and *Parageobacillus*.

5. In line 234-235, the sentence should be revised.

Our response:

Thanks for your comment. We have revised the sentence in the new version.

6. In Supplementary Figure 4, the description about 4B and 4C was absent in the Result section.

Our response:

Thank you for your comment.

We have addressed this issue to ensure that the text aligns with the figures presented in the revised version.

7. In Figure 5D, it may be better to represent enriched tRNAs as the relative read count fold change compared to initial competent cell library in Figure 5C, rather than the number of read count, to avoid the bias generated from the initial plasmid library or the competent cell library.

Our response:

Thank you for your insightful suggestion.

We agree with you that representing enriched tRNAs as the relative read count fold change compared to the initial competent cell library in Figure 5C is a more appropriate approach. This modification will help mitigate any potential bias arising from the initial plasmid library or the competent cell library.

We have revised Figure 5D as you suggested.

8. In line 402-417, the repression efficiency of each five tgRNA in the multiplexing array according to their order may be tested.

Our response:

Thank you for your suggestion.

We agree that the order-dependent effects of tgRNAs in multiplexing arrays.

To ensure the best combination effect of five tgRNA, we arranged the tgRNAs based on their attributions to transformation efficiency (Fig. 5E and 5F). Our evaluations confirmed that this order of the five tgRNAs represents the optimal arrangement.

9. In line 450-466, please clarify how genome-scale CRISPRi library was introduced to Gt-08 strain which seems to already have multiplex repression CRISPRi vector (five genes for enhancing competency).

Our response:

Thank you for your comments.

In Figure 5F, we depicted the construction of a set of tgRNAs specifically targeting the five transformation-related target genes. The tgRNA array was then introduced into the genome of GtCom6 to create strain GtCom7, which served as the initial strain for riboflavin production. It's important to note that the strain Gt-08 (GtCom7 derivant with 5 targets knockout, 2 targets knock-in, and 2 targets point mutation, Figure 6A) itself did not harbored plasmid. So the genome-scale CRISPRi library was able to introduce the cell.

10. In line 450-466, are the identified target genes by CRISPRi for enhancing riboflavin titers could be predicted by the genome-scale metabolic model (<https://doi.org/10.1016/j.ymben.2021.03.002>)?

Our response:

Thank you for your comments.

To answer this question, we utilized the genome-scale metabolic model (GEM) mentioned in your provided reference¹² (<https://doi.org/10.1016/j.ymben.2021.03.002>) along with the modified method¹³ (<https://doi.org/10.1073/pnas.2108245119>) to predicted the possible targets. However, we found that these predicted targets (Additional Table 1) did not match our identified targets by CRISPRi.

Our CRISPRi screening, which was only performed one round of screening as a proof of concept, resulted in the identification of 10 positive targets. Among these 10 positive targets, 4 were either regulatory protein or unannotated protein, which inherently cannot be predicted by GEM (only containing 1175 reactions). This may explain the discrepancy between the predicted results and the outcome of our CRISPRi screening.

Additional table 1

Gene ID	Reaction name in model *	Reaction equation
RTMO02835 or RTMO02836	Acetyl-CoA:formate C-acetyltransferase	Acetyl-CoA + Formate \rightleftharpoons CoA + Pyruvate
RTMO02228	Isocitrate:NADP ⁺ oxidoreductase (decarboxylating)	Isocitrate + NADP ⁺ \rightleftharpoons CO ₂ + 2-Oxoglutarate + NADPH
RTMO01831 or RTMO01473	(S)-malate:NAD ⁺ oxidoreductase	CO ₂ + Pyruvate + NADH \rightleftharpoons (S)-Malate + NAD ⁺
RTMO02229	acetyl-CoA:oxaloacetate C-acetyltransferase (thioester-hydrolysing)	CoA + Citrate + H ⁺ \rightleftharpoons Water + Acetyl-CoA + Oxaloacetate
RTMO04156 or RTMO04574	Succinate:CoA ligase (ADP-forming)	CoA + ATP + Succinate \rightleftharpoons Succinyl-CoA + ADP + Orthophosphate
RTMO04978 or RTMO04977 or RTMO04976	succinate:quinone oxidoreductase	Menaquinone + Succinate \rightleftharpoons Menaquinol + Fumarate
RTMO02253	GTP:pyruvate 2-O-phosphotransferase	Pyruvate + GTP \rightleftharpoons GDP + Phosphoenolpyruvate
RTMO00678	formate:NAD ⁺ oxidoreductase	NAD ⁺ + Formate \rightarrow CO ₂ + NADH
RTMO02318	2-phospho-D-glycerate hydro-lyase (phosphoenolpyruvate-forming)	2-Phospho-D-glycerate \rightleftharpoons Water + Phosphoenolpyruvate
RTMO02253	UTP:pyruvate 2-O-phosphotransferase	Pyruvate + UTP \rightleftharpoons UDP + Phosphoenolpyruvate
RTMO01048	D-glucosamine-6-phosphate ketol-isomerase(deaminating)	Water + D-Glucosamine \rightleftharpoons Ammonium + D-Fructose 6-phosphate
RTMO02321	D-glyceraldehyde-3-phosphate aldose-ketose-isomerase	Glyceronephosphate \rightleftharpoons Glyceraldehyde 3-phosphate
RTMO05021	D-glyceraldehyde-3-phosphate:NADP ⁺ oxidoreductase (phosphorylating)	Glyceraldehyde 3-phosphate + NADP ⁺ + Orthophosphate \rightleftharpoons 3-Phospho-D-glyceroyl + NADPH
RTMO03967	D-fructose-1,6-bisphosphate D-glyceraldehyde-3-phosphate-lyase (glycerone-phosphate-forming)	D-Fructose \rightleftharpoons Glyceraldehyde 3-phosphate + Glyceronephosphate
RTMO00614	(S)-malate hydro-lyase (fumarate-forming)	Water + Fumarate \rightleftharpoons (S)-Malate
RTMO03752	citrate hydro-lyase (cis-aconitate-forming)	Water + cis-Aconitate \rightleftharpoons Citrate
RTMO02322	ATP:3-phospho-D-glycerate 1-phosphotransferase	3-Phospho-D-glycerate + ATP + H ⁺ \rightleftharpoons 3-Phospho-D-glyceroyl + ADP
RTMO02320	D-phosphoglycerate 2,3-phosphomutase	2-Phospho-D-glycerate \rightleftharpoons 3-Phospho-D-glycerate
RTMO03752	isocitrate hydro-lyase (cis-aconitate-forming)	Isocitrate \rightleftharpoons Water + cis-Aconitate
RTMO03238	D-glucosamine 1,6-phosphomutase	D-Glucosamine \rightleftharpoons alpha-D-Glucosamine
RTMO05024	ATP:D-fructose-6-phosphate 1-phosphotransferase	ATP + D-Fructose 6-phosphate \rightarrow D-Fructose + ADP
RTMO04263	Selenocystathionine Lysteine-lyase (deaminating)	Water + L-Selenocystathionine \rightarrow L-Selenocysteine + Ammonium + 2-Oxobutanoate
RTMO03881	O-phosphorylhomoserine phosphate-lyase (adding selenocysteine)	L-Selenocysteine + O-Phospho-L-homoserine \rightarrow L-Selenocystathionine + H ⁺ + Orthophosphate
RTMO00874 or RTMO00873 or RTMO00135 or RTMO00827 or RTMO01152	2-Oxoglutarate dehydrogenase complex	CoA + 2-Oxoglutarate + NAD ⁺ \rightarrow CO ₂ + Succinyl-CoA + NADH
RTMO05960	Transport of metabolite o2_e	C00007 \rightleftharpoons Oxygen

RTMO05961	Transport of metabolite co2_e	CO2 <=>C00011
RTMO06049	ATP synthase (four protons for one ATP)	ADP + Orthophosphate + 4 Proton <=>Water + ATP + 3 H+
RTMO06053	NADH Dehydrogenase (ubiquinone & 3.5 protons)	Menaquinone + NADH + 4.5 H+ ->Menaquinol + NAD+ + 3.5 Proton
RTMO06054	Cytochrome oxidase bo3 (ubiquinol: 2.5 protons)	Menaquinol + 0.5 Oxygen + 2.5 H+ ->Water + Menaquinone + 2.5 Proton
RTMO06109	Transport of L-Arginine	Water + ATP + L-Arginine ->L-Arginine + H+ + ADP + Orthophosphate
RTMO06109	Transport of L-Arginine	L-Arginine + H+ <=>Proton + L-Arginine
RTMO06170	Pimelate-Coa:ACP transferase	Acyl-carrier + Pimeloyl-CoA <=>CoA + 4 H+ + Pimeloyl-[acyl-carrier protein]
RTMO06185	glyceraldehyde-3-phosphate dehydrogenase	Glyceraldehyde 3-phosphate + NAD+ + Orthophosphate ->3-Phospho-D-glyceroyl + NADH

11. A previous report (<https://doi.org/10.1021/acssynbio.1c00138>) in the same species developed thermostable CRISPR-Cas9 system. Please emphasize or provide the evidence for the novelty or efficiency of the type I-B CRISPR-Cas system of this study compared to the Cas9 system, in addition to the advantages of PAM described in discussion section (line 493-497).

Our response:

Thank you for your comments.

Here we listed another three points to show the novelty or efficiency of the type I-B CRISPR-Cas system compared to the Cas9 system:

- (1) The type I-B CRISPR-Cas system had remarkable thermostability (20–70 °C), whereas the stCas9 system functioned at 52°C in *P. thermoglucosidasius*. So, the thermostability of this type I-B CRISPR-Cas system is better, which allows this system broader application environments that require robust protein activity.
- (2) Given *P. thermoglucosidasius* grows much faster at 60°C compared that at 52°C (the temperature stCas9 worked), the type I-B CRISPR-Cas system must complete editing within less time at 60°C (a life cycle of the host). So, we believe the activity of this type I-B CRISPR-Cas system is higher.
- (3) Type I-B system is orthogonal, allowing for the switching between genome editing and transcriptional repression activities by control crRNA length without ablating Cas3 nuclease; also allowing synchronous genome editing and transcriptional repression.

12. In line 557, “thespacer” should be “the spacer”.

Our response:

Thank you for your careful review.

We have corrected the mistake in line 557.

13. In Supplementary Note 2, please describe the design principle of tgRNA library, including rationale for avoiding off-targets. How was the top three tgRNAs selected for each gene?

What are the noteworthy results?

Our response:

Thank you for your comments.

Based on the results shown in Supplementary Figure 8, we observed that tgRNAs targeting the template strand exhibited stronger suppression compared to those targeting the non-template strand. Moreover, tgRNAs targeting upstream, midstream, or downstream regions of the template strand demonstrated similar repression activity.

Considering these findings, we designed the tgRNA library using the following principles:

1. We chose tgRNA from non-template strand.
2. we selected 3 tgRNAs from upstream, midstream, and downstream regions of each gene.

Since we have confirmed that the type I-B CRISPR-Cas system exhibited highly specific transcriptional repression without significant off-target effects (Figure 3F), we did not consider off-target effect of tgRNA here.

Reference

1. Xu F, *et al.* Characterization of a Riboflavin-Producing Mutant of *Bacillus subtilis* Isolated by Droplet-Based Microfluidics Screening. *Microorganisms* **11**, 1070 (2023).
2. Wagner JM, Liu L, Yuan S-F, Venkataraman MV, Abate AR, Alper HS. A comparative analysis of single cell and droplet-based FACS for improving production phenotypes: Riboflavin overproduction in *Yarrowia lipolytica*. *Metabolic engineering* **47**, 346-356 (2018).
3. Li M, *et al.* Toxin-antitoxin RNA pairs safeguard CRISPR-Cas systems. *Science* **372**, eabe5601 (2021).
4. Du K, Gong L, Li M, Yu H, Xiang H. Reprogramming the endogenous type I CRISPR - Cas system for simultaneous gene regulation and editing in *Haloarcula hispanica*. *mLife* **1**, 40-50 (2022).

5. Alerasool N, Segal D, Lee H, Taipale M. An efficient KRAB domain for CRISPRi applications in human cells. *Nature Methods* **17**, 1093-1096 (2020).
6. Konermann S, *et al.* Genome-scale transcriptional activation by an engineered CRISPR-Cas9 complex. *Nature* **517**, 583-588 (2015).
7. Wang T, *et al.* Pooled CRISPR interference screening enables genome-scale functional genomics study in bacteria with superior performance. *Nature communications* **9**, 2475 (2018).
8. Martínez-Espinosa RM. Heterologous and homologous expression of proteins from Haloarchaea: Denitrification as case of study. *Int J Mol Sci* **21**, 82 (2019).
9. Walker JE, *et al.* Development of both type I–B and type II CRISPR/Cas genome editing systems in the cellulolytic bacterium *Clostridium thermocellum*. *Metabolic engineering communications* **10**, e00116 (2020).
10. Sheng L, Kovács K, Winzer K, Zhang Y, Minton NP. Development and implementation of rapid metabolic engineering tools for chemical and fuel production in *Geobacillus thermoglucosidasius* NCIMB 11955. *Biotechnology for Biofuels* **10**, 1-18 (2017).
11. Yang Z, *et al.* Engineering thermophilic *Geobacillus thermoglucosidasius* for riboflavin production. *Microbial Biotechnology* **14**, 363-373 (2021).
12. Mol V, *et al.* Genome-scale metabolic modeling of *P. thermoglucosidasius* NCIMB 11955 reveals metabolic bottlenecks in anaerobic metabolism. *Metabolic Engineering* **65**, 123-134 (2021).
13. Ishchuk OP, *et al.* Genome-scale modeling drives 70-fold improvement of intracellular heme production in *Saccharomyces cerevisiae*. *Proceedings of the National Academy of Sciences* **119**, e2108245119 (2022).

Reviewers' Comments:

Reviewer #1:

Remarks to the Author:

Authors addressed all the issues.

Suggest publication.

Reviewer #2:

Remarks to the Author:

The authors put generally adequate effort to respond to the provided comments. They also show that their system is not toxic and has no off-targets effects, in contrast to the currently characterised 3 thermophilic Cas9 orthologs. Nonetheless, my penultimate comment still lacks adequate referencing. One of the provided references (#45) is a review article that generally refers to the toxicity and off-targets of some Cas9 orthologs, but does not specifically include any study on the 3 thermophilic Cas9. In the other provided reference (#46) the 'toxicity assays were used to demonstrate functional DNA cutting for both the Type I-B and Type II systems'; hence the authors of that study used 'toxicity' and 'targeted DNA cleaving' interchangeably.

I would suggest that the authors either provide new accurate references to support their claim, that they specifically talk about the thermophilic Cas9s, or rephrase lines 498-501 (original manuscript) generalizing their statement for Cas9 orthologs.

Reviewer #3:

Remarks to the Author:

* Comments for Author:

The reviewer would like to express gratitude for all the comments and provide sincere responses of the authors. It is believed that their responses and revisions have further improved the manuscript. However, there are a few additional comments for which they would appreciate receiving responses.

Comments

1. For the response of previous comment 1, it was understood that proteins from halophilic organisms are characterized by their high content of acidic amino acids (Aspartate and glutamic acid), being only stable in solutions containing high salt concentration. However, there is still curiosity about the reason of the broad stability (due to thermodynamically stable fold) of type I-B CRISPR-Cas of *P. thermoglucosidasius* in both thermophile and mesophile. Is it also common in other Cas effectors or other proteins of thermophiles? But the authors explain in the response of previous comment 11, stCas9 didn't show thermostability in broad range.
2. For the response of previous comment 8, please provide the confirmation data as the Supplementary Figure that the array order (12375-09555-01915-08150-09535) was the optimal arrangement.
3. In Figure 2F, changing the color of the control graph to white may be better for unity with 1C-1E.
4. In Figure 2G, changing the color of the control graph to white may be better for unity with 1C-1E. Also, statistical significance may be indicated.
5. In Line 462-465, Supplementary Figure 12A, Table 6, and Supplementary Figure 12B may be capital.
6. In Line 542, MutS gene was one of the downregulation target for increasing transformation efficiency. However, the repression of this gene may increase the mutation rate, which may affect negatively as the production chassis. Please describe about this aspect.

Point-by-point response

Dear editor and reviewers,

We greatly appreciate the constructive comments and suggestions, which are valuable and helpful for improving our manuscript. We have fully addressed all the comments, and listed the point-by-point response as following.

Reviewer #1 (Remarks to the Author):

Authors addressed all the issues.

Suggest publication.

Reviewer #2 (Remarks to the Author):

The authors put generally adequate effort to respond to the provided comments. They also show that their system is not toxic and has no off-targets effects, in contrast to the currently characterised 3 thermophilic Cas9 orthologs. Nonetheless, my penultimate comment still lacks adequate referencing. One of the provided references (#45) is a review article that generally refers to the toxicity and off-targets of some Cas9 orthologs, but does not specifically include any study on the 3 thermophilic Cas9. In the other provided reference (#46) the 'toxicity assays were used to demonstrate functional DNA cutting for both the Type I - B and Type II systems'; hence the authors of that study used 'toxicity' and 'targeted DNA cleaving' interchangeably.

I would suggest that the authors either provide new accurate references to support their claim, that they specifically talk about the thermophilic Cas9s, or rephrase lines 498-501 (original manuscript) generalizing their statement for Cas9 orthologs.

Our response:

Thanks for your kind suggestion.

We agree with you that the current studies on toxicity and off-target effects of these thermophilic Cas9s are still rare. So, we rephrased lines 498-501 as you suggested.

We previously discussed the toxicity and off-target effects arose from our own experiments. We found that ThermoCas9¹ was difficult to transform in to *P. thermoglucosidasius* and was ineffective for target cleavage in this heterologous strain. Given those data was not sufficient and not closely related to this work, we here rephrased lines 498-501 as you suggested.

Reviewer #3 (Remarks to the Author):

* Comments for Author:

The reviewer would like to express gratitude for all the comments and provide sincere responses of the authors. It is believed that their responses and revisions have further

improved the manuscript. However, there are a few additional comments for which they would appreciate receiving responses.

Comments

1. For the response of previous comment 1, it was understood that proteins from halophilic organisms are characterized by their high content of acidic amino acids (Aspartate and glutamic acid), being only stable in solutions containing high salt concentration. However, there is still curious about the reason of the broad stability (due to thermodynamically stable fold) of type I-B CRISPR-Cas of *P. thermoglucosidasius* in both thermophile and mesophile. Is it also common in other Cas effectors or other proteins of thermophiles? But the authors explain in the response of previous comment 11, stCas9 didn't showed thermostability in broad range.

Our response:

Thanks for your comment.

The activity test indicated that this type I-B CRISPR-Cas worked well at 37–70 °C *in vivo*. We are also curious about the underlying reason of the broad stability, so we are preparing this type I-B complex and analyzing the structure by electron microscopy now. We agree with you that the thermodynamically stable fold and interaction confer this complex the broad stability (Additional Figure 1 and Table 1). This work is ongoing now.

Based on many crystal structures of the reported thermophilic enzymes, several factors that are responsible for thermostability have been illustrated, such as a selection of hydrophobic cores^{2,3}, buried polar contacts and ion pairs⁴, amino acid substitutions⁵, and interactions among subunits^{6,7}.

As for whether other Cas effectors or other proteins of thermophiles show broad stability, to our knowledge, there is no systematic study. Generally, some enzymes used for genetic engineering are from thermophilic organisms and do also tolerate the lower temperature, such as DNA polymerases used in PCR and many endonucleases used in genetic manipulation. The thermoCas9-based engineering tools also have demonstrated the applications of gene deletion and transcriptional silencing at 55 °C in *Bacillus smithii* and of gene deletion at 37 °C in *Pseudomonas putida*¹.

In previous comment 11, we aimed to provide the evidence for the efficiency of the type I-B CRISPR-Cas, and we did not deny that stCas9 showed thermostability. As for stCas9 system functioned at 52 °C in *P. thermoglucosidasius*, we ascribed to the probable cause that the optimal temperature of its native host *Streptococcus thermophilus* dose not reach 50 °C⁸. Likewise, previous studies have demonstrated that spCas9 is inactive at temperatures above 42 °C due to the optimal temperature of its native host is 37 °C⁹.

Additional Figure 1. Purification and thermal analysis of *P. thermoglucosidasius* Cascade complexes including His-Cas7 and 38 bp mature crRNA. The cascade complex was purified by metal-affinity chromatography and size-exclusion chromatography. Subsequently, the purified complex was incubated at 50°C, 65°C, 80°C, and 95°C for 20 min. The supernatant of complex after centrifugation were then analyzed using a 4–20% SDS-polyacrylamide gel. Molecular weight markers are represented in lane M, while the Cascade subunits are indicated. This result indicated that this purified type I-B complex was stable enough to be incubated at 65°C or below this temperature. While at 80 and 95 °C, the type I-B complex was precipitated completely and unable to be detected in the supernatant.

Additional Table 1. Mass-spectrometry analysis of Cascade-crRNA complex obtained by PAGE gel. This result confirmed each component of the type I-B complex.

Protein	Mass (Da)	Score	Coverage (%)	Peptides
Can7	35531	35947	98.1	AIFLSATR MFGATMPK LFKEYDGK SLNIVSDVR MFGATMPK YEGATNPK LLEVNVEAR VDGETVNATER NDGGNGKQVR DQWLEETEGLR NNSDLRYDAK MFGATMPKSEDK GLADLPSLADK SSSFTLVKDALVNR VEYNSPFIAGDILR DYLEQGHFVFGK LTNPNGDIDEENPR LTEDRHIFDQAVYK EQGNWMLDQFIVR VYVFLAFHVGSGHR VTVESASITSHFGSDDK YRDIYEQGHFVFGK RHYVQGSNLTNTEGK LSKEDQNWMLDQFIVR GSSMTTGFQFNWVSYLNK DRYVYVSLAHGVISGHR ITRFDHIFDQAVVKAIFLSATR VTVESASITSHFGSDKNDDGNGK SRDKSSMTTGFQFNWVSYLNK SISEVTLHVDALVNLLEVNVEIAR DYLEQGHFVFGKVDGETVNATER VEYNSPFIAGDILRHLLEETEGLR MNNSDLRYDAKLTNPNGDIDEENPR MLFPLDGK VMAQTVNMFVK LSNLVPVHSVVR KVMQVTVNMFVK TTVGMAGLGEMER IDTNSSLSVAFPPR KIDTNSLSVAFPPR MICEIDVLEKDSR QJAVAVPVAVAVGEAK DSYVEVSPDQQIASVR AHLNNSGGHTQIPELVLPQGEANLR VYVPPLEGLSELISGLQZVAEAGMLK TVNKPILSK ISFASOLEK LSVFEDAKK LFDLALQEK EAPVGETTK MLQNLIK AEALGGAWGK ISFASOLEK TVETAEDLHR DVTFDLGAAK LLSQVPLHLR ARALGGAWGK LKFNVINDK ILQVYESLTKK MLQNLNKEER KILQVYESLTKK LMLLSHEVFDKLR WDDQKQVSYLK LMLLSHEVFDKLR IASLQNETVIEHK DANQAMAGLAVLR IASLQNETVIEHKR FNGLFHYLPMR SLVQWLTQVQLAK YVFNPAVYSGELTEGK FNFTYNDKGFASOLEK YVFNPAVYSGELTEGK YLTEEKPSMQAITLPIAK VHMVAVNLYVYSOKEELPNDR EANVNIKDELVYSAINQQLVVSQPEYER LESEK NSQSGMPEK FQAFQAPTEK THFSPDAEFSK RTHFSPDAEFSK MLSPLTVSYTENQHGK INAWNGMYEQAPLPVYK SIQDFELAHFDVGYTK INAWNGMYEQAPLPVYK NLSPLTVSYTENQHGK NGALIPNYELVLOGFLYR TGEIGQR KGATQVR INHWGAPK TVNKPILSK LFDLALQEK FAPVGETTK AEALGGAWGK MLQNLIK TVETAEDLHR DVTFDLGAAK RLPDALQEK LLSQVPLHLR ARALGGAWGK LMLLSHEVFDK KDNSQDFENEE MLQNLNKEER LMLLSHEVFDKLR QRILMILLSHEVFDK IASLQNETVIEHK ESVPYLLSQVAGTK IASLQNETVIEHKR ESVPYLLSQVAGTKR YVFNPAVYSGELTEGK AEALGGAWGKTVNKPILSK YVFNPAVYSGELTEGK YLTEEKPSMQAITLPIAK WNLSDRSPVYLLSQVAGTK VTYDLDHAMVETMETMNVNVSQGLLQVLAAR
Can5	28152	3039	80.7	VMAQTVNMFVK LSNLVPVHSVVR KVMQVTVNMFVK TTVGMAGLGEMER IDTNSSLSVAFPPR KIDTNSLSVAFPPR MICEIDVLEKDSR QJAVAVPVAVAVGEAK DSYVEVSPDQQIASVR AHLNNSGGHTQIPELVLPQGEANLR VYVPPLEGLSELISGLQZVAEAGMLK TVNKPILSK ISFASOLEK LSVFEDAKK LFDLALQEK EAPVGETTK MLQNLIK AEALGGAWGK ISFASOLEK TVETAEDLHR DVTFDLGAAK LLSQVPLHLR ARALGGAWGK LKFNVINDK ILQVYESLTKK MLQNLNKEER KILQVYESLTKK LMLLSHEVFDKLR WDDQKQVSYLK LMLLSHEVFDKLR IASLQNETVIEHK DANQAMAGLAVLR IASLQNETVIEHKR FNGLFHYLPMR SLVQWLTQVQLAK YVFNPAVYSGELTEGK FNFTYNDKGFASOLEK YVFNPAVYSGELTEGK YLTEEKPSMQAITLPIAK VHMVAVNLYVYSOKEELPNDR EANVNIKDELVYSAINQQLVVSQPEYER LESEK NSQSGMPEK FQAFQAPTEK THFSPDAEFSK RTHFSPDAEFSK MLSPLTVSYTENQHGK INAWNGMYEQAPLPVYK SIQDFELAHFDVGYTK INAWNGMYEQAPLPVYK NLSPLTVSYTENQHGK NGALIPNYELVLOGFLYR TGEIGQR KGATQVR INHWGAPK TVNKPILSK LFDLALQEK FAPVGETTK AEALGGAWGK MLQNLIK TVETAEDLHR DVTFDLGAAK RLPDALQEK LLSQVPLHLR ARALGGAWGK LMLLSHEVFDK KDNSQDFENEE MLQNLNKEER LMLLSHEVFDKLR QRILMILLSHEVFDK IASLQNETVIEHK ESVPYLLSQVAGTK IASLQNETVIEHKR ESVPYLLSQVAGTKR YVFNPAVYSGELTEGK AEALGGAWGKTVNKPILSK YVFNPAVYSGELTEGK YLTEEKPSMQAITLPIAK WNLSDRSPVYLLSQVAGTK VTYDLDHAMVETMETMNVNVSQGLLQVLAAR
Can8	75825	3009	58	VMAQTVNMFVK LSNLVPVHSVVR KVMQVTVNMFVK TTVGMAGLGEMER IDTNSSLSVAFPPR KIDTNSLSVAFPPR MICEIDVLEKDSR QJAVAVPVAVAVGEAK DSYVEVSPDQQIASVR AHLNNSGGHTQIPELVLPQGEANLR VYVPPLEGLSELISGLQZVAEAGMLK TVNKPILSK ISFASOLEK LSVFEDAKK LFDLALQEK EAPVGETTK MLQNLIK AEALGGAWGK ISFASOLEK TVETAEDLHR DVTFDLGAAK LLSQVPLHLR ARALGGAWGK LKFNVINDK ILQVYESLTKK MLQNLNKEER KILQVYESLTKK LMLLSHEVFDKLR WDDQKQVSYLK LMLLSHEVFDKLR IASLQNETVIEHK DANQAMAGLAVLR IASLQNETVIEHKR FNGLFHYLPMR SLVQWLTQVQLAK YVFNPAVYSGELTEGK FNFTYNDKGFASOLEK YVFNPAVYSGELTEGK YLTEEKPSMQAITLPIAK VHMVAVNLYVYSOKEELPNDR EANVNIKDELVYSAINQQLVVSQPEYER LESEK NSQSGMPEK FQAFQAPTEK THFSPDAEFSK RTHFSPDAEFSK MLSPLTVSYTENQHGK INAWNGMYEQAPLPVYK SIQDFELAHFDVGYTK INAWNGMYEQAPLPVYK NLSPLTVSYTENQHGK NGALIPNYELVLOGFLYR TGEIGQR KGATQVR INHWGAPK TVNKPILSK LFDLALQEK FAPVGETTK AEALGGAWGK MLQNLIK TVETAEDLHR DVTFDLGAAK RLPDALQEK LLSQVPLHLR ARALGGAWGK LMLLSHEVFDK KDNSQDFENEE MLQNLNKEER LMLLSHEVFDKLR QRILMILLSHEVFDK IASLQNETVIEHK ESVPYLLSQVAGTK IASLQNETVIEHKR ESVPYLLSQVAGTKR YVFNPAVYSGELTEGK AEALGGAWGKTVNKPILSK YVFNPAVYSGELTEGK YLTEEKPSMQAITLPIAK WNLSDRSPVYLLSQVAGTK VTYDLDHAMVETMETMNVNVSQGLLQVLAAR
Can6	29444	1124	66	VMAQTVNMFVK LSNLVPVHSVVR KVMQVTVNMFVK TTVGMAGLGEMER IDTNSSLSVAFPPR KIDTNSLSVAFPPR MICEIDVLEKDSR QJAVAVPVAVAVGEAK DSYVEVSPDQQIASVR AHLNNSGGHTQIPELVLPQGEANLR VYVPPLEGLSELISGLQZVAEAGMLK TVNKPILSK ISFASOLEK LSVFEDAKK LFDLALQEK EAPVGETTK MLQNLIK AEALGGAWGK ISFASOLEK TVETAEDLHR DVTFDLGAAK LLSQVPLHLR ARALGGAWGK LKFNVINDK ILQVYESLTKK MLQNLNKEER KILQVYESLTKK LMLLSHEVFDKLR WDDQKQVSYLK LMLLSHEVFDKLR IASLQNETVIEHK DANQAMAGLAVLR IASLQNETVIEHKR FNGLFHYLPMR SLVQWLTQVQLAK YVFNPAVYSGELTEGK FNFTYNDKGFASOLEK YVFNPAVYSGELTEGK YLTEEKPSMQAITLPIAK VHMVAVNLYVYSOKEELPNDR EANVNIKDELVYSAINQQLVVSQPEYER LESEK NSQSGMPEK FQAFQAPTEK THFSPDAEFSK RTHFSPDAEFSK MLSPLTVSYTENQHGK INAWNGMYEQAPLPVYK SIQDFELAHFDVGYTK INAWNGMYEQAPLPVYK NLSPLTVSYTENQHGK NGALIPNYELVLOGFLYR TGEIGQR KGATQVR INHWGAPK TVNKPILSK LFDLALQEK FAPVGETTK AEALGGAWGK MLQNLIK TVETAEDLHR DVTFDLGAAK RLPDALQEK LLSQVPLHLR ARALGGAWGK LMLLSHEVFDK KDNSQDFENEE MLQNLNKEER LMLLSHEVFDKLR QRILMILLSHEVFDK IASLQNETVIEHK ESVPYLLSQVAGTK IASLQNETVIEHKR ESVPYLLSQVAGTKR YVFNPAVYSGELTEGK AEALGGAWGKTVNKPILSK YVFNPAVYSGELTEGK YLTEEKPSMQAITLPIAK WNLSDRSPVYLLSQVAGTK VTYDLDHAMVETMETMNVNVSQGLLQVLAAR
Can8b	75825	17092	43.8	VMAQTVNMFVK LSNLVPVHSVVR KVMQVTVNMFVK TTVGMAGLGEMER IDTNSSLSVAFPPR KIDTNSLSVAFPPR MICEIDVLEKDSR QJAVAVPVAVAVGEAK DSYVEVSPDQQIASVR AHLNNSGGHTQIPELVLPQGEANLR VYVPPLEGLSELISGLQZVAEAGMLK TVNKPILSK ISFASOLEK LSVFEDAKK LFDLALQEK EAPVGETTK MLQNLIK AEALGGAWGK ISFASOLEK TVETAEDLHR DVTFDLGAAK LLSQVPLHLR ARALGGAWGK LKFNVINDK ILQVYESLTKK MLQNLNKEER KILQVYESLTKK LMLLSHEVFDKLR WDDQKQVSYLK LMLLSHEVFDKLR IASLQNETVIEHK DANQAMAGLAVLR IASLQNETVIEHKR FNGLFHYLPMR SLVQWLTQVQLAK YVFNPAVYSGELTEGK FNFTYNDKGFASOLEK YVFNPAVYSGELTEGK YLTEEKPSMQAITLPIAK VHMVAVNLYVYSOKEELPNDR EANVNIKDELVYSAINQQLVVSQPEYER LESEK NSQSGMPEK FQAFQAPTEK THFSPDAEFSK RTHFSPDAEFSK MLSPLTVSYTENQHGK INAWNGMYEQAPLPVYK SIQDFELAHFDVGYTK INAWNGMYEQAPLPVYK NLSPLTVSYTENQHGK NGALIPNYELVLOGFLYR TGEIGQR KGATQVR INHWGAPK TVNKPILSK LFDLALQEK FAPVGETTK AEALGGAWGK MLQNLIK TVETAEDLHR DVTFDLGAAK RLPDALQEK LLSQVPLHLR ARALGGAWGK LMLLSHEVFDK KDNSQDFENEE MLQNLNKEER LMLLSHEVFDKLR QRILMILLSHEVFDK IASLQNETVIEHK ESVPYLLSQVAGTK IASLQNETVIEHKR ESVPYLLSQVAGTKR YVFNPAVYSGELTEGK AEALGGAWGKTVNKPILSK YVFNPAVYSGELTEGK YLTEEKPSMQAITLPIAK WNLSDRSPVYLLSQVAGTK VTYDLDHAMVETMETMNVNVSQGLLQVLAAR

2. For the response of previous comment 8, please provide the confirmation data as the Supplementary Figure that the array order (12375-09555-01915-08150-09535) was the optimal arrangement.

Our response:

Thanks for your interest.

As for the array order (12375-09555-01915-08150-09535), we initially arranged the tgRNAs based on their respective attributions to transformation efficiency (Fig. 5E and 5F). We presumed that the such order of the five tgRNAs represents the optimal arrangement, because the proximal of CRISPR array usually be transcribed more efficiently¹⁰.

To substantiate this presumption, we conducted tests by comparing four random array orders with the first design. The results indicated that the first design indeed was the optimal arrangement among the five tested array orders (see Additional Figure 2). Additionally, we observed that different array orders did not have much significant influence on the transformation efficiency. So, here we did not test more combinations. For another reason, there are a total of 120 possible combinations in theory. Constructing and testing all combinations was extremely challenging. we, therefore in the previous version, did not provide the confirmation data.

Now, we have added this information in the Supplementary Figure 10H.

Additional Figure 2. Transformation efficiency of strains with different array orders. The blue column represents the optimal arrangement, while the yellow columns denote the four alternative array orders.

3. In Figure 2F, changing the color of the control graph to white may be better for unity with 1C-1E.

Our response:

Thanks for your comment. We have revised as you suggested.

4. In Figure 2G, changing the color of the control graph to white may be better for unity with 1C-1E. Also, statistical significance may be indicated.

Our response:

Thanks for your comment. We have revised as you suggested.

5. In Line 462-465, Supplementary Figure 12A, Table 6, and Supplementary Figure 12B may be capital.

Our response:

Thanks for your comment. We have revised as you suggested.

6. In Line 542, MutS gene was one of the downregulation target for increasing transformation efficiency. However, the repression of this gene may increase the mutation rate, which may affect negatively as the production chassis. Please describe about this aspect.

Our response:

Thanks for your comment.

MutS–MutL-based canonical mismatch repair (MMR) system is highly conserved between prokaryotes and eukaryotes. In bacteria loss of the mismatch repair capacity results in increasing spontaneous mutation frequencies. Additionally, it is reported that deletion of MutS increased the transformation efficiency in several strains^{11, 12, 13, 14}. Combining the identification MutS as one of the downregulation target for increasing transformation efficiency in our work, we speculated that MutS might play multiple roles *in vivo*.

To avoid the increase of mutation rate, we here dynamically controlled the repression of these targets by a xylose-induced promoter, not deleted simplistically. Only when performing genetic transformation, the inducer is added to the medium to turn on the competent state instantaneously. In medium without the inducer the cells remain healthy and metabolically active. Thus, the use of an inducible promoter to drive the repression of multiple targets results in a dynamic switch that preserves cellular integrity.

Reference:

1. Mougiakos I, *et al.* Characterizing a thermostable Cas9 for bacterial genome editing and silencing. *Nature communications* **8**, 1647 (2017).
2. Bezudnova EY, *et al.* Structural insight into the molecular basis of polyextremophilicity of short-chain alcohol dehydrogenase from the

- hyperthermophilic archaeon *Thermococcus sibiricus*. *Biochimie* **94**, 2628-2638 (2012).
3. Chen J, Stites WE. Replacement of staphylococcal nuclease hydrophobic core residues with those from thermophilic homologues indicates packing is improved in some thermostable proteins. *J Mol Biol* **344**, 271-280 (2004).
 4. Hakulinen N, Turunen O, Jänis J, Leisola M, Rouvinen J. Three-dimensional structures of thermophilic beta-1,4-xylanases from *Chaetomium thermophilum* and *Nonomuraea flexuosa*. Comparison of twelve xylanases in relation to their thermal stability. *European journal of biochemistry / FEBS* **270**, 1399-1412 (2003).
 5. Óskarsson KR, Sævarsson AF, Kristjánsson MM. Thermostabilization of VPR, a kinetically stable cold adapted subtilase, via multiple proline substitutions into surface loops. *Scientific reports* **10**, 1045 (2020).
 6. Nakka M, Iyer RB, Bachas LG. Intersubunit disulfide interactions play a critical role in maintaining the thermostability of glucose-6-phosphate dehydrogenase from the hyperthermophilic bacterium *Aquifex aeolicus*. *The protein journal* **25**, 17-21 (2006).
 7. Pang J, Allemann RK. Molecular dynamics simulation of thermal unfolding of *Thermatoga maritima* DHFR. *Physical chemistry chemical physics : PCCP* **9**, 711-718 (2007).
 8. Lau MSH, Sheng LL, Zhang Y, Minton NP. Development of a Suite of Tools for Genome Editing in *Parageobacillus thermoglucosidasius* and Their Use to Identify the Potential of a Native Plasmid in the Generation of Stable Engineered Strains. *Acs Synthetic Biology* **10**, 1739-1749 (2021).
 9. Mougiakos I, *et al.* Efficient Genome Editing of a Facultative Thermophile Using Mesophilic spCas9. *ACS Synth Biol* **6**, 849-861 (2017).
 10. Yang J, Han YH, Im J, Seo SW. Synthetic protein quality control to enhance full-length translation in bacteria. *Nat Chem Biol* **17**, 421-427 (2021).
 11. Meier P, Wackernagel W. Impact of mutS inactivation on foreign DNA acquisition by natural transformation in *Pseudomonas stutzeri*. *J Bacteriol* **187**, 143-154 (2005).
 12. Majewski J, Cohan FM. The effect of mismatch repair and heteroduplex formation on sexual isolation in *Bacillus*. *Genetics* **148**, 13-18 (1998).
 13. Majewski J, Zawadzki P, Pickerill P, Cohan FM, Dowson CG. Barriers to genetic exchange between bacterial species: *Streptococcus pneumoniae* transformation. *J Bacteriol* **182**, 1016-1023 (2000).

14. Zhou H, Zhang L, Xu Q, Zhang L, Yu Y, Hua X. The mismatch repair system (mutS and mutL) in *Acinetobacter baylyi* ADP1. *BMC microbiology* **20**, 40 (2020).

Reviewers' Comments:

Reviewer #3:

Remarks to the Author:

The authors provide point-by-point responses for all additional comments with sufficient data and rationale. Particularly, it was intriguing that the authors are now analyzing the structure of the type I-B Cas complex as follow-up research. The outcomes of the follow-up research would greatly enhance the understanding of thermostable proteins.

Point-by-point response

Dear editor and reviewers,

We greatly appreciate the constructive comments and suggestions, which are valuable and helpful for improving our manuscript. We have fully addressed all the comments, and listed the point-by-point response as following.

Reviewer #3 (Remarks to the Author):

The authors provide point-by-point responses for all additional comments with sufficient data and rationale. Particularly, it was intriguing that the authors are now analyzing the structure of the type I-B Cas complex as follow-up research. The outcomes of the follow-up research would greatly enhance the understanding of thermostable proteins.

Our response:

Thank you for your comments. We appreciate your interest in our follow-up research on the structure of the type I-B Cas complex.